# Spatial rearrangement of the *Streptomyces venezuelae* linear chromosome during sporogenic development

Marcin J. Szafran[1✉], Tomasz Małecki[1], Agnieszka Strzałka[1], Katarzyna Pawlikiewicz[1], Julia Duława[1], Anna Zarek[1], Agnieszka Kois-Ostrowska[1], Kim C. Findlay [2], Tung B. K. Le [2] & Dagmara Jakimowicz [1✉]

Bacteria of the genus *Streptomyces* have a linear chromosome, with a core region and two 'arms'. During their complex life cycle, these bacteria develop multi-genomic hyphae that differentiate into chains of exospores that carry a single copy of the genome. Sporulation-associated cell division requires chromosome segregation and compaction. Here, we show that the arms of *Streptomyces venezuelae* chromosomes are spatially separated at entry to sporulation, but during sporogenic cell division they are closely aligned with the core region. Arm proximity is imposed by segregation protein ParB and condensin SMC. Moreover, the chromosomal terminal regions are organized into distinct domains by the *Streptomyces*-specific HU-family protein HupS. Thus, as seen in eukaryotes, there is substantial chromosomal remodelling during the *Streptomyces* life cycle, with the chromosome undergoing rearrangements from an 'open' to a 'closed' conformation.

[1] Faculty of Biotechnology, University of Wrocław, Wrocław, Poland. [2] The John Innes Centre, Norwich Research Park, Norwich, UK.
✉email: marcin.szafran@uwr.edu.pl; dagmara.jakimowicz@uwr.edu.pl

Chromosomal organisation differs notably among the domains of life, and even among bacteria, the chromosomal arrangement is not uniform. While linear eukaryotic chromosomes undergo dramatic condensation before their segregation, the compaction of usually circular bacterial chromosomes remains relatively constant throughout the cell cycle[1–3]. To date, chromosome conformation capture methods have been applied in only a few model bacterial species, revealing that bacterial chromosomes adopt one of two distinct conformation patterns. Either both halves of the circular chromosome (replichores) align closely (as in *Bacillus subtilis*[4,5], *Caulobacter crescentus*[6], *Corynebacterium glutamicum*[7], *Mycoplasma pneumoniae*[8], and chromosome II of *Vibrio cholerae*[9]) or they tend to adopt an 'open' conformation, which lacks close contacts between both chromosomal arms (as in *Escherichia coli*[10] and *V. cholerae* chromosome I[9]). Cytological studies have shown that chromosome organisation may alter depending on the bacterial growth phase or developmental stage[1,11–13]. While the most prominent changes in bacterial chromosome organisation can be observed in the extensive DNA compaction that occurs during sporulation of *Bacillus* spp.[14] and *Streptomyces* spp.[13], the chromosome conformation during this process has not been elucidated.

The folding of bacterial genomes requires a strictly controlled interplay between several factors, including macromolecular crowding, transcription and the activity of various dedicated DNA-organising proteins (topoisomerases, condensins and nucleoid-associated proteins (NAPs))[10,15,16]. By non-specific DNA binding, NAPs induce its bending, bridging or wrapping to govern the local organisation and global compaction of bacterial chromosomes[1,10,16]. Since the subcellular repertoire of NAPs is dependent on growth conditions, changes in their supply can account for the fluctuation of chromosome compaction in response to environmental clues. Furthermore, the repertoire of NAPs also varies between bacterial species, with some (such as HU homologues) being widespread while others are less conserved[13,17,18]. By contrast, condensins (SMC-ScpAB complex found in most phyla, or the MukBEF complex found in the gammaproteobacteria including *E. coli*) are among the most conserved DNA-organising proteins identified in all domains of life[19–21]. According to a recently proposed model, the primary activity of SMC is the extrusion of large DNA loops[22,23]. SMC complexes were shown to move along bacterial chromosomes from the centrally positioned origin of replication (*oriC*), leading to gradual chromosome compaction and parallel alignment of both replichores[5,6,24,25]. Importantly, although the alignment of two chromosomal arms was not seen in *E. coli*, the MukBEF complex is still responsible for long-range DNA interactions[10,26].

In *B. subtilis*, *C. glutamicum* and *C. crescentus*, the loading of SMC in the vicinity of the *oriC* were shown to be dependent on the chromosome segregation protein ParB[4,6,7,27,28]. Binding to a number of *parS* sequences, ParB forms a large nucleoprotein complex (segrosome), that interacts with ParA, an ATPase bound non-specifically to nucleoid[29,30]. Interaction with ParB triggers ATP hydrolysis and nucleoid release by ParA, generating a concentration gradient that drives segrosome separation[31–33]. Segrosome formation was shown to be a prerequisite for chromosomal arm alignment in Gram-positive bacteria with circular chromosomes[5,7,28]. There have been no reports on the 3D organisation of linear bacterial chromosomes to date.

*Streptomyces* spp. are among a few bacterial genera that possess linear chromosome. They are Gram-positive mycelial soil actinobacteria, which are highly valued producers of antibiotics and other clinically important metabolites. Notably, the production of these secondary metabolites is closely correlated with *Streptomyces* unique and complex life cycle, which encompasses vegetative growth and sporulation, as well as exploratory growth in some species[34,35]. During the vegetative stage, *Streptomyces* grow as branched hyphae composed of elongated cell compartments, each containing multiple copies of nonseparated and largely uncondensed chromosomes[1,3,36]. Under stress conditions, sporogenic hyphae develop, and their rapid elongation is accompanied by extensive replication of chromosomes[37]. When hyphae stop extending, the multiple chromosomes undergo massive compaction and segregation to generate unigenomic spores[13]. These processes are accompanied by multiple synchronised cell divisions initiated by FtsZ, which assembles into a ladder of regularly spaced Z-rings[38,39]. Chromosome segregation is driven by ParA and ParB proteins, whereas nucleoid compaction is induced by the concerted action of SMC and sporulation-specific NAPs that include HupS - one of the two *Streptomyces* HU homologues, structurally unique to actinobacteria owing to the presence of a histone-like domain[40–44]. Thus, while in unicellular bacteria chromosome segregation occurs during ongoing replication, during *Streptomyces* sporulation, intensive chromosome replication is temporally separated from their compaction and segregation, with the latter two processes occurring only during sporogenic cell division.

While *Streptomyces* chromosomes are among the largest bacterial chromosomes, our knowledge concerning their organisation is scarce. These linear chromosomes are flanked with terminal inverted repeats (TIRs) which encompass palindromes that form telomeric secondary structures. Replication of telomeres is mediated by a covalently bound telomere-associated complex of terminal proteins (TPs)[45,46]. The elimination of TPs results in circularisation of the chromosome[47,48]. Interestingly, the length of TIRs varies for different *Streptomyces* species, ranging from less than a few hundred nucleotides (in *S. avermitilis*[49]), to 200 kbp (in *S. ambofaciens*[50]) or even over 600 kbp (in *S. collinus*[51]), while in *S. venezuelae* it was determined to be about 1.5 kbp[52]. Limited cytological evidence suggests that during vegetative growth, both ends of the linear chromosome colocalize, and the *oriC* region of the apical chromosome is positioned at the tip-proximal edge of the nucleoid[53,54]. The visualisation of the nucleoid and ParB-marked *oriC* regions in sporogenic hyphae showed that *oriC*s are positioned centrally within pre-spore compartments, but did not elucidate chromosome architecture during sporulation[39,54].

In this study, we investigated if *S. venezuelae* chromosome organisation changes during sporulation. Using the chromosome conformation capture method (Hi-C), we observed a dramatic rearrangement of chromosome structure associated with sporogenic development. Moreover, by combining the Hi-C with chromatin immunoprecipitation (ChIP) and fluorescence microscopy techniques, we investigated the contribution of ParB, SMC and HupS (the sporulation-dedicated HU homologue) to global nucleoid compaction. We demonstrated the substantial rearrangement of the *S. venezuelae* chromosomes which undertake compact folded conformation during sporogenic cell division.

## Results

**Hi-C reveals alignment of chromosomal arms extending to terminal domains during *S. venezuelae* sporogenic cell division.** During *Streptomyces* sporulation, multi-genomic hyphae undergo considerable morphological changes, turning into a chains of spores through multiple cell divisions. This event is linked to the inhibition of DNA replication followed by synchronised compaction and segregation of tens of chromosomes[37,39,55–57]. To explore whether the increasing chromosome compaction during sporulation is reflected in overall changes in chromosome organisation, we optimised Hi-C[6,58,59] to map DNA contact frequency across the *S. venezuelae* chromosome (Fig. 1 and Supplementary Fig. 1).

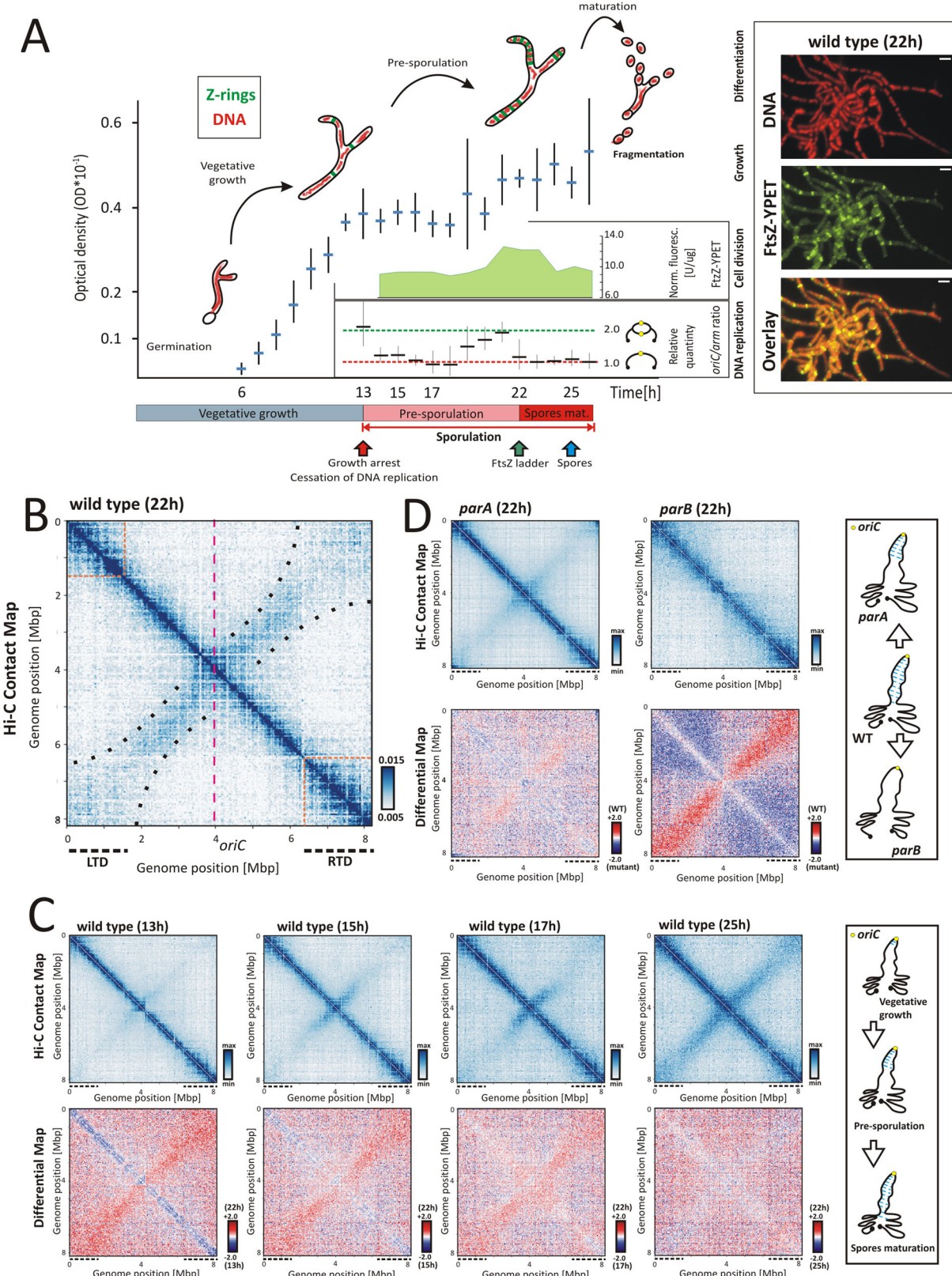

First, using *ftsZ-ypet* strain, and having confirmed that production of FtsZ-Ypet fusion protein does not affect *S. venezuelae* growth (Supplementary Fig. 2A), we determined the time points at which critical sporulation events took place, namely growth arrest and cessation of DNA replication, formation of the FtsZ ladder, and appearance of spores (Fig. 1A). After 13–14 h of

growth (under 5 ml culture conditions), we observed vegetative growth inhibition. Slowed hyphal growth correlated with a reduction in the *oriC/arm* ratio corresponding to decreased DNA replication (Fig. 1A and Supplementary Fig. 3A), indicating entry into a developmental stage that we named the pre-sporulation phase, and which lasted until 22 h of growth (~8 h).

**Fig. 1 Spatial organisation of the *S. venezuelae* linear chromosome during sporulation. A** The *S. venezuelae* (wild-type *ftsZ-ypet* derivative, MD100) growth curve (in 5 ml culture), performed in $n = 3$ independent experiments. The mean optical density values and standard deviation (whiskers) are shown on the scheme. The critical time points of sporogenic development are marked with arrows: growth arrest and cessation of DNA replication (red), the appearance of FtsZ ladders (green) and the spore formation (blue). The normalised FtsZ-YPet fluorescence (a marker of a synchronous cell division) [U/μg] and the relative *oriC*/arm ratio (a marker of DNA replication; $n = 3$ independent experiments: mean *oriC*/arm values as well as calculated standard deviations (whiskers) are shown on the diagram) are shown as insets in the plot, with X axes corresponding to the main plot X axis. The *oriC*/arm ratio at the 26 h of growth was set as 1.0. The right panel shows the representative visualisation of condensing nucleoids ($n = 250$ analysed nucleoids per a single time point; DNA stained with 7-AAD) and FtsZ-YPet at 22 h of growth; scale bar 2 μm. **B** The normalised Hi-C contact map obtained for the wild-type (*ftsZ-ypet* derivative, MD100) after 22 h of growth (in 5 ml culture). The dotted pink lines mark the position of the *oriC* region. The dotted black lines mark the positions of the left (LTD) and right (RTD) terminal domains. The boundaries of the LTD and RTD are marked directly on the Hi-C contact map with the orange dotted lines. The contact range within the secondary diagonal axis is marked with black dots. **C** Top panel: the normalised Hi-C contact maps obtained for the wild-type (*ftsZ-ypet* derivative, MD100) strain growing for 13, 15, 17 and 25 h (in 5 ml culture). Bottom panel: the differential Hi-C maps in the logarithmic scale (log2) comparing the contact enrichment at 22 h of growth (red) versus 13, 15, 17 and 25 h of growth (blue). The X and Y axes indicate chromosomal coordinates binned in 30 kbp. The right panel shows the model of chromosome rearrangement in the course of sporulation. **D** Top panel: the normalised Hi-C contact maps generated for *parA* and *parB* mutants (*ftsZ-ypet* derivatives MD011 and MD021, respectively) growing for 22 h (in 5 ml culture). Bottom panel: the differential Hi-C maps in the logarithmic scale (log2) comparing the contact enrichment in the wild-type strain (red) versus the mutant strain (blue). The right panel shows the model of chromosome organisation in the *parA* and *parB* mutants.

During this phase, the DNA replication rate transiently increased (in 19–21 h), and this increased replication rate coincided with a slight increase in the optical density of the culture, which could be explained by the rapid growth of sporogenic hyphae (Fig. 1A). A subsequent drop in the replication rate was accompanied by an increase in the FtsZ-YPet level (21–23 h) and assembly of FtsZ-YPet into ladders of Z-rings (22 h) (Supplementary Fig. 4A). At this time point, the spore chains entered the maturation phase; during this phase, chromosome compaction was shown to gradually increase, achieving significant condensation at 25 h (Supplementary Fig. 4B), while the FtsZ-YPet level rapidly decreased and the hyphae fragmented. This stage ended when immature spores could be detected (25–26 h) (Fig. 1A and Supplementary Fig. 4A). Thus, we established that 22 h of *S. venezuelae* culture is the point at which multi-genomic hyphae undergo synchronised cell division to form spores, replication of chromosomes ceases, while their condensation increases, which would be expected to minimise interchromosomal contacts. Based on these findings, we set out to establish the chromosome conformation at this time point.

Hi-C mapping of the chromosomal contacts in *S. venezuelae* sporogenic hyphae (22 h) undergoing cell division indicated the presence of two diagonal axes crossing perpendicularly in the proximity of the centrally located *oriC* region (Fig. 1B). The primary diagonal corresponds to the contact frequency between neighbouring chromosomal loci. Based on the contact range (extended in the proximity of chromosome termini) and principal component analysis (PCA1), three distinct chromosome regions were distinguished: the core region that encompasses 4.4 Mbp (position: 1.9–6.3 Mbp) around the centrally positioned *oriC* and two well-organised domains flanking the chromosome core, which we named the left and right terminal domains (LTD and RTD, respectively), with each measuring ~1.5–2 Mbp in size (Fig. 1B and Supplementary Fig. 5). The secondary diagonal, which results from the interarm interactions, was prominent in the region of ~2.0–2.2 Mbp in each direction from the *oriC* region and severely diminished within terminal domains (Fig. 1B). Additionally, the Hi-C contact map showed some diffuse signals at the termini proximal fragments of the secondary diagonal, suggesting the existence of spatial proximity but also somewhat longer range contacts at the termini of the linear *S. venezuelae* chromosome (Fig. 1B and Supplementary Fig. 3B).

In summary, these results indicate that at the time of *S. venezuelae* synchronised cell division, the linear chromosomes are folded with both arms closely aligned within the core region, while the terminal regions form distinct domains.

**Arms proximity increases during sporulation and requires segregation protein activity.** Having established chromosomal contact maps during *S. venezuelae* sporogenic cell division, we set out to characterise the changes in chromosomal arrangement during sporogenic development. To this end, we obtained normalised Hi-C contact maps for the wild-type strain during the vegetative stage at the entry into the pre-sporulation phase when growth slows and DNA replication decreases (13 h) and during the pre-sporulation phase when the chromosomes are still not compacted (15 and 17 h), as well as during spore maturation (25 h), which is when chromosome compaction reaches a maximum (Fig. 1A and Supplementary Fig. 4B).

At the entry into the pre-sporulation phase (13 h), only the primary diagonal was clearly visible, whereas the secondary diagonal was barely detectable across the core and terminal domains, indicating limited contact between the chromosomal arms during vegetative growth (Fig. 1C). During the pre-sporulation phase, the signals of the secondary diagonal were enhanced and extended at distances of up to 1 Mbp from the *oriC* region at 15 and 17 h of growth (Fig. 1C). Finally, in the maturing spores (25 h), we observed almost complete arm alignment, similar to that detected at 22 h (Fig. 1C). This result suggests that although nucleoid compaction was still underway during the spores maturation phase (Supplementary Fig. 4B), the global chromosome organisation did not change critically after that established during cell division (Fig. 1C). Since the contact maps show various chromosomal arrangements within the hyphae of *S. venezuelae*, we cannot estimate the timing of the process for a single chromosome. However, taking into account synchronous *S. venezuelae* development, we can conclude that the average arm cohesion gradually increases during the pre-sporulation phase.

In model bacteria whose chromosomal arms are in close proximity, their positioning is directed by ParB-dependent SMC loading within the *oriC* region[4,7,28]. During *Streptomyces* sporogenic cell division, coincident with the time of observed folding of chromosomes with proximal arms, ParB has been shown to bind numerous *parS* scattered in the proximity of *oriC* (Supplementary Fig. 3B), assembling into regularly spaced complexes that position *oriC* regions along the hyphae[57]. At the same time, ParA accumulates in sporogenic hyphae and facilitates the efficient formation and segregation of ParB complexes[39,60]. Elimination of *S. venezuelae* segregation proteins disturbs aerial hyphal development and chromosome segregation[39] but does not affect the culture growth rate (Supplementary Fig. 2B). To verify whether segregation proteins have a role in the organisation of *S. venezuelae*

chromosomes, we obtained Hi-C maps of chromosomal contacts in sporogenic hyphae of *parA* or *parB* mutants.

The Hi-C map generated for the *parB* mutant showed the complete disappearance of the secondary diagonal axis, confirming the expected role of the ParB complex in facilitating interarm contacts (Fig. 1D). The same lack of a secondary diagonal was observed for the *parB* mutant at each time point in the *S. venezuelae* life cycle (Supplementary Fig. 6A). Deletion of *parA* also had a detectable influence on arm alignment, although it varied among the samples analysed (Fig. 1D and Supplementary Fig. 6B). The strength of the signals at the secondary diagonal was either significantly or only slightly lowered in the *parA* mutant, as indicated by the differential Hi-C map in comparison to the wild-type strain (Fig. 1D and Supplementary Fig. 6B). While the effect of *parA* or *parB* deletions was predominantly manifested by the partial or complete disappearance of the interarm contacts, respectively, their influence on the local chromosome structure, including LTD and RTD folding, was marginal (Fig. 1D and Supplementary Fig. 5).

Taken together, these results indicate that chromosomal arm alignment progresses during pre-sporogenic and sporogenic hyphal development (Fig. 1C, scheme) and that this chromosomal rearrangement is dependent on ParB (Fig. 1D, scheme).

### SMC- and HupS-induced chromosome compaction permits regularly spaced cell division.

In many bacteria, ParB impacts chromosome organisation by mediating the loading of SMC, which promotes global chromosome compaction. Prior studies using *S. coelicolor* showed that the elimination of either SMC or HupS, a *Streptomyces*-specific sporulation-associated NAP, affected nucleoid compaction in spores[40,42,61]. We hypothesised that SMC would be responsible for ParB-induced chromosomal arm alignment but that since ParB elimination did not influence short-range chromosome interactions, other DNA-organising factors, such as HupS, may also contribute to chromosome compaction in sporogenic hyphae.

The disruption of either the *hupS* or *smc* gene in *S. venezuelae* did not significantly affect the growth rate or differentiation of these bacteria; however, a double *hupS smc* mutant grew slower than wild-type or either of the single mutants (Supplementary Fig. 7A, inset and Supplementary Fig. 7B). Interestingly, the elimination of *hupS* or, to a lesser extent, *smc* increased spore size, while the double *hupS smc* deletion led to more pronounced disturbances in spore size (Fig. 2A, B). Normal spore length was restored in the single mutants by complementation with in trans-delivered *hupS-FLAG* or *smc-FLAG* genes (Fig. 2B).

Next, to visualise nucleoid compaction and segregation during cell division, we used fluorescence microscopy to observe sporogenic hyphal development of *smc* and/or *hupS* mutant strains labelled with HupA-mCherry and FtsZ-YPet (Fig. 2C and Supplementary Fig. 7C, Supplementary Movies 1–4). None of the mutant strains formed anucleate spores, but all of them showed significantly increased nucleoid areas, in particular the double *hupS smc* mutant, where notably enlarged spores could be observed (Fig. 2A–D). The increased nucleoid area was accompanied by increased distances between Z-rings (Supplementary Fig. 7D). The largest fraction was found for Z-rings spaced at increased distances (longer than 1.3 μm) in the double *hupS smc* mutant and in the *smc* mutant (Supplementary Fig. 7D). Additionally, the intensity of Z-rings in single *smc* or the double mutant was more varied along the hyphae than in the wild-type strain, indicating their lowered stability (Fig. 2E, F). The disturbed positioning and stability of Z-rings explain the presence of elongated spores in mutant strains.

In summary, both SMC and HupS contribute to chromosome compaction in spores. The decreased chromosome compaction in the mutant strains correlated with decreased Z-ring stability and their aberrant positioning which, in all studied mutants, resulted in increased distance between septa and elongated spores.

### SMC and HupS collaborate during sporogenic chromosome rearrangement.

Having confirmed that both SMC and HupS mediate global chromosome compaction during *S. venezuelae* sporulation, we investigated how both proteins contribute to chromosomal contacts during sporulation-associated chromosome rearrangement.

The Hi-C contact map obtained for the *smc* mutant at sporogenic cell division (22 h) showed the complete disappearance of the secondary diagonal (similar to the map obtained for the *parB* mutant), indicating abolished chromosomal arm interactions (Figs. 1D and 3A). In contrast, the elimination of *smc* resulted in an increase in short-range DNA contact frequency (<200 kbp) along the primary diagonal, whereas in the *parB* mutant, the short-range DNA contacts were only slightly affected, suggesting an additional role of SMCs in chromosome organisation (Figs. 1D, 3A and Supplementary Fig. 8A). Indeed, while nucleoid compaction was previously noted to be affected by *parB* deletion[39], the measurements of the nucleoid area confirmed higher chromosome compaction in *parB* than in *smc* mutant strains (Supplementary Fig. 8B).

To confirm that ParB promotes the interaction of SMC with DNA, we used ChIP-seq to analyse FLAG-SMC binding in the wild-type background and in the *parB* mutant. First we assessed how the developmental time points for 50-ml culture conditions used in our ChIP-seq experiments corresponded to those in the 5 ml culture previously described for Hi-C experiments (Fig. 1A and Supplementary Fig. 9A). At the 12 h of 'ChIP-seq culture', DNA replication decreased (corresponding to 15–17 h of 'Hi-C culture'), while at 14 h of 'ChIP-seq culture', DNA compaction and the formation of regularly spaced Z-rings were clearly detected (Supplementary Fig. 9B) (corresponding to 22 h in 'Hi-C culture' conditions), indicating sporogenic cell division and chromosome segregation. Notably, the level of FLAG-SMC remained constant during sporogenic development (Supplementary Fig. 10A) and the N-terminal FLAG-SMC fusion did not affect the function of the protein (Supplementary Fig. 10B).

Analysis of DNA fragments bound by FLAG-SMC at the time of sporogenic cell division did not identify any specific binding sites for FLAG-SMC, suggesting a lack of sequence preference. However, quantification of the detected SMC binding sites along the chromosome (using an algorithm dedicated to non-specific DNA binders) showed that their frequency increased up to twofold in the chromosomal core region (Fig. 3B and Supplementary Fig. 11A). The increased frequency of SMC binding overlapped with the secondary diagonal identified in Hi-C experiments (Fig. 3A, B and Supplementary Fig. 11B). SMC binding was severely diminished in the *parB* mutant (Fig. 3B), demonstrating that the ParB complex is a prerequisite for SMC loading.

The Hi-C contact map generated for the *hupS* mutant at the time sporogenic cell division (at the 22 h) revealed the presence of both diagonals. However, the identified local signals within both diagonals were more dispersed than in the wild-type, and diagonals were visibly broadened (Fig. 3A and Supplementary Fig. 6C). Moreover, along the primary diagonal, a less clear distinction between the core region and terminal domains could be observed. The effect of *hupS* deletion, while spreading over the

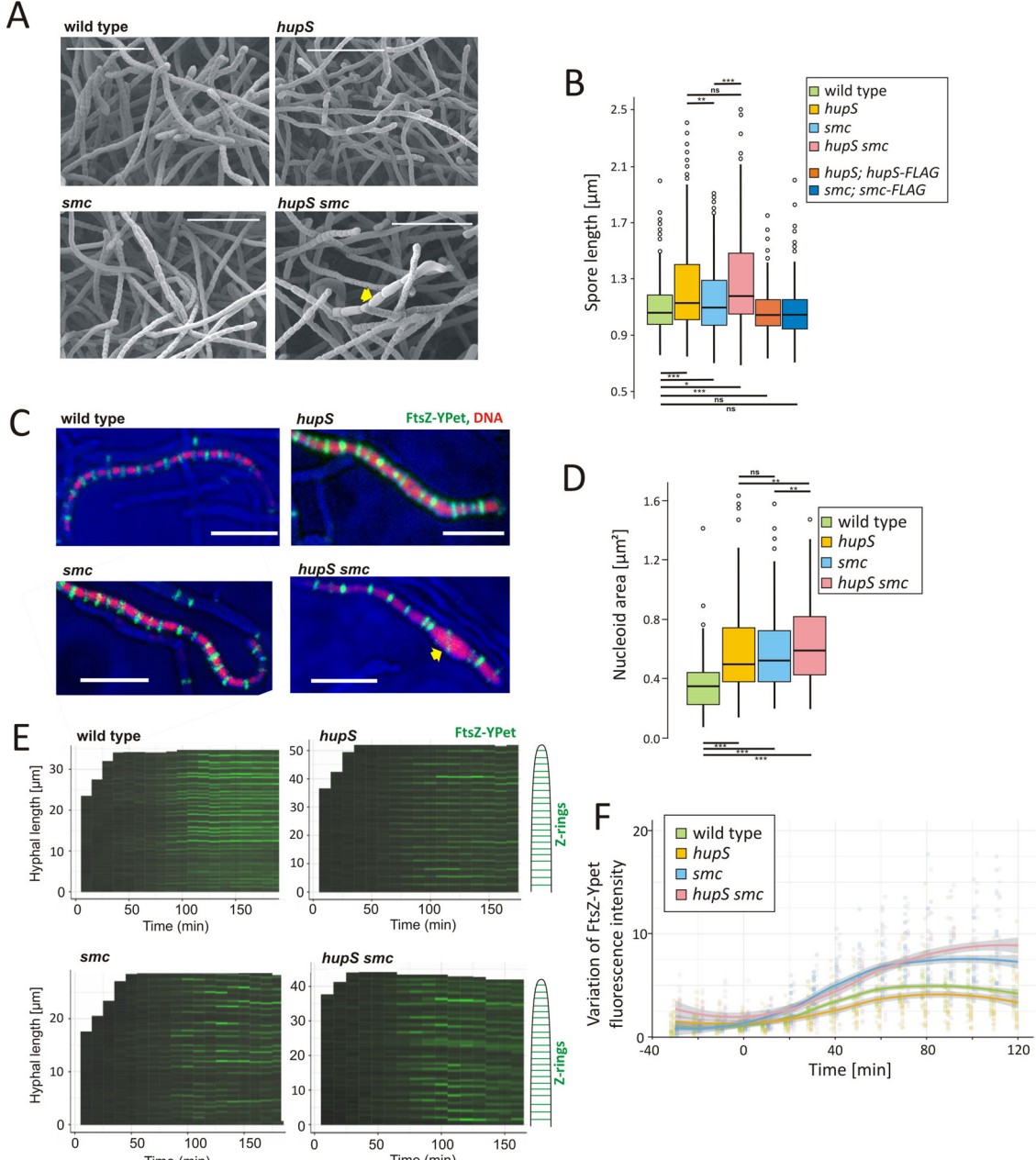

whole chromosome compaction, was most prominent within terminal domains, especially within RTD (Fig. 3A and Supplementary Fig. 5).

To further investigate the involvement of HupS in the organisation of terminal domains and to test the cooperation between HupS and SMC, we analysed HupS-FLAG binding in the complemented *hupS* and double *hupS smc* deletion strains at the time of cell division. In contrast to SMC, HupS-FLAG showed low sequence binding preference, and we identified 307 regions differentially bound by HupS-FLAG with a mean width of 221 bp (Fig. 3C, D and Supplementary Fig. 12A). The analysis of the number of HupS-FLAG binding sites exhibited a significant increase in the terminal domains, especially the right terminal domain, but HupS binding was also detected in the core region of the chromosome (Fig. 3C). The enhanced binding of HupS observed at chromosomal termini is consistent with the lowered number of chromosomal contacts in RTD observed in the Hi-C maps obtained for the *hupS* mutant.

Further analysis of ChIP-seq data performed using two independent methods (*rGADEM* and *MEME*) identified the motif recognised by HupS-FLAG (Supplementary Fig. 12B). Although the motif was observed to be uniformly widespread along the chromosome, and was not required for HupS binding, it was enriched in HupS binding sites versus random regions, with an increased probability for occurrence in the middle of the HupS-bound regions (confirmed with *CentriMo*) (Supplementary Figs. 12B–E). Furthermore, in vitro studies showed that immobilised GST-HupS-bound digested cosmid DNA with no sequence specificity (Supplementary Fig. 12F). Interestingly, ChIP-seq analysis of HupS-FLAG binding in the absence of SMC (*hupS smc* deletion complemented with *hupS*-FLAG) showed a lower number of reads for all HupS-FLAG binding sites, indicating that SMC enhanced the HupS-DNA interaction (Fig. 3D, E). This observation could not be explained by the lowered level of HupS-FLAG in the *smc* deletion background (Supplementary Fig. 10C). We also noted marginal effect of *hupS*

**Fig. 2 Phenotypic effects of *smc*, *hupS*, and double *smc/hupS* deletions. A** Scanning electron micrographs showing sporulating hyphae and spores of the wild-type and *hupS*, *smc* and *hupS smc* double mutant (*ftsZ-ypet* derivatives, MD100, TM005, TM004 and TM006, respectively) (representative of 10 images). Elongated spores are marked with a yellow arrowhead. Scale bar: 10 μm. **B** Box plot analysis of the spore length distribution in the *S. venezuelae* wild-type (light green), *hupS* (yellow), *smc* (blue) and *hupS smc* (pink) (AK200, TM010, TM003, respectively), as well as in *hupS* and *smc* mutants complemented with in trans-delivered *hupS-flag* (orange) and *smc-flag* genes (dark blue) (TM015 and TM019, respectively) (300 spores were analysed for each strain). Boxplots show median with first and third quartile while the lower and upper 'whiskers' extend to the value no further than 1.5 * IQR (interquartile range) from the 'hinge'. The statistical significance between strains determined by a one-way anova with a Games–Howell post-hoc test (two-sided) is marked with asterisks: *p*-value ≤0.05 (*), ≤0.01 (**) and ≤0.001 (***). *p*-values: wild-type-*hupS*: <0.0001, wild-type-*smc*: 0.0242, wild-type-*hupS smc*: <0.0001, wild-type-*hupS*; *hupS-FLAG*: 0.0625, wild-type-*smc*; *smc-FLAG*: 0.3871, *hupS-smc*: 0.0089, *hupS-hupS smc*: 0.1241, *smc-hupS smc*: <0.0001. **C** Examples of time-lapse images (representative of 10 repetitions) showing visualisation of nucleoids (marked with HupA-mCherry fusion) and Z-rings (visualised by FtsZ-YPet fusion) in the wild-type background (*ftsZ-ypet*, *hupA-mCherry* derivative, TM011) *hupS*, *smc* and *hupS smc* double mutant background (*ftsZ-ypet*, *hupA-mCherry* derivative, TM013, TM012 and TM014, respectively). An abnormal hyphal fragment is marked with a yellow arrowhead. Scale bar: 5 μm. **D** Box plot analysis of the nucleoid area (visualised by HupA-mCherry fusion) in the wild-type (light green, 262 spores), *hupS* (yellow, 232 spores), *smc* (blue, 220 spores) and *hupS smc* (pink, 218 spores) double mutant (*ftsZ-ypet*, *hupA-mcherry* derivative, MD100, TM011, TM013, TM012, TM014, respectively). Boxplots show median with first and third quartile while the lower and upper 'whiskers' extend to the value no further than 1.5 * IQR (interquartile range) from the 'hinge'. The statistical significance between strains determined by Wilcoxon rank-sum test (two-sided) with Holm method used for multiple comparisons is marked with asterisks: *p*-value ≤0.05 (*), ≤0.01 (**) and ≤0.001 (***). *p*-values: wild-type-*hupS*: <2e−16, wild-type-*smc*: <2e−16, wild-type-*hupS smc*: <2e−16, *hupS-smc*: 0.9082, *hupS-hupS smc*: 0.0028, *smc-hupS smc*: 0.0039. **E** Kymographs showing the intensity of Z-ring fluorescence along the representative hyphae in the wild-type and *hupS*, *smc* and *hupS smc* double mutant (*ftsZ-ypet* derivatives, MD100, TM005, TM004, TM006, respectively) during maturation. **F** The variation (standard deviation) of FtsZ-Ypet (Z-rings) fluorescence during spore maturation of strains calculated for 25–30 hyphae of the wild-type (light green), *hupS* (yellow), *smc* (blue) and *hupS smc* (pink) double mutant (*ftsZ-ypet* derivatives, TM005, TM004, TM006, respectively). Fluorescence was measured starting from 30 min before growth arrest until spore formation. Points represent values for each hyphae, while lines show the result of Local Polynomial Regression Fitting (loess) with 95% confidence intervals (shaded area).

deletion on chromosome organisation at the early pre-sporulation phase, as indicated by the Hi-C contact map for the *hupS* mutant at this time point (Supplementary Fig. 10D), which corresponded with the lower HupS-FLAG level during the vegetative growth than during sporulation[42].

Since our ChIP-seq analysis indicated cooperation between HupS and SMC binding and that the double *hupS smc* deletion most severely affected chromosome compaction, we investigated how the double deletion affects chromosomal arrangement. The Hi-C contact map generated for the double *hupS smc* mutant at the time of sporogenic cell division (22 h) showed the combination of effects observed independently for *hupS* and *smc* single mutations, namely, the disruption of arm alignment and the destabilization of local DNA contacts (Fig. 3A). Thus, while long-range interarm interactions are facilitated by SMC, short-range compaction is governed by HupS, and the effect of the elimination of both proteins is additive.

In summary, our Hi-C and ChIP-seq analyses of *smc* and *hupS* mutants showed their contribution to global chromosome compaction. While SMC binds in the chromosomal core and is responsible for the close contact of chromosome arms, HupS binds preferentially in terminal domains and organises them. Cooperation between both proteins is manifested by SMC enhancing HupS binding to DNA. Elimination of both proteins disturbs global chromosome organisation.

## Discussion

In this study, we provide a complex picture of the dynamic rearrangement of a linear bacterial chromosome. Hi-C mapping of the ~8.2 Mbp *S. venezuelae* chromosome during sporogenic cell division revealed chromosomal folding with interarm contacts, which extend ~4 Mbp around the *oriC* region. Arms proximity is diminished at the independently folded 1.5 and 2 Mbp terminal domains LTD and RTD, respectively (Fig. 4). The independent folding of core and terminal domain corroborates with distinguished by genomic analyses regions of *Streptomyces* chromosome: core that contains housekeeping genes and exhibits high synteny and 1.5–2 Mbp long arms with low evolutionary conservation[62]. Further analysis of chromosomal contacts shows spatial proximity of both chromosomal termini, which is

consistent with the findings of previous cytological studies[53]. Folding of the *Streptomyces* linear chromosome into core and terminal domains was also independently demonstrated for the *S. ambofaciens*[63].

Our Hi-C data disclosed substantial rearrangement of chromosomal conformation occurring during *S. venezuelae* sporulation. At the end of vegetative growth, at the entrance to the sporulation phase contacts between the chromosomal arms are scarce. After this time point, chromosomal arms become increasingly aligned, reaching a maximum at the time of initiation of sporogenic cell division (Z-ring formation) which is also the time point when chromosome segregation is initiated[39]. Thus, during *Streptomyces* differentiation, changes in hyphal morphology, gene expression, chromosome replication and mode of cell division[13,34] are accompanied by rearrangement of chromosomes from an extended conformation during vegetative growth to almost fully folded and compacted structures.

Our data demonstrate that in *S. venezuelae*, both ParB and SMC are essential for imposing chromosomal arms proximity, reinforcing previous work describing ParB-dependent SMC loading onto the chromosome. Importantly, in *S. venezuelae*, SMC binding is enriched within the core region of the chromosome, matching the higher frequency of interarm contacts in this region. SMC was previously shown to be involved in chromosome compaction during *S. coelicolor* sporulation[40,61]. Interestingly, *parB* deletion was also observed to decrease chromosome condensation at the time of cell division[39]. This observation may be explained by the lack of SMC loading in absence of ParB, which was also reported for the other model bacteria[4,6,7,9]. The increase in long-range chromosomal contacts induced by SMC in the chromosomes of *S. venezuelae* and other bacteria corroborates the proposed model of action in which SMC zips DNA regions when translocating from the loading site[4].

While the increase in arm alignment during sporulation[63] cannot be explained by increased SMC level during sporulation, the *parAB* genes were both earlier shown to be transcriptionally induced in *Streptomyces* sporogenic hyphae[56,64]. ParB binds in the proximity of the *oriC* regions of all chromosomes throughout the whole life cycle but complexes formed during vegetative growth and at the early stages of sporogenic development are not

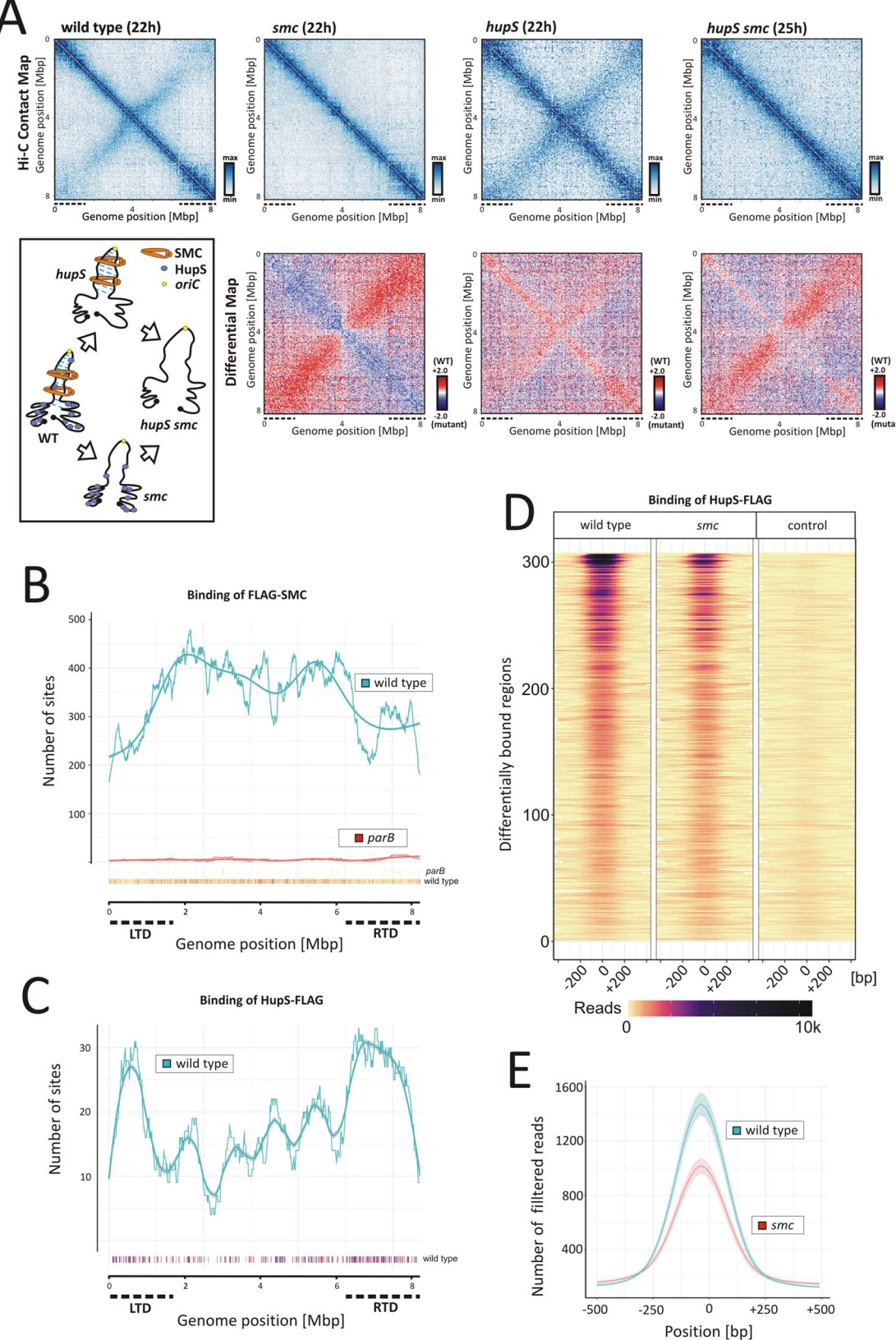

as massive as the regularly spaced segrosomes that accompany sporogenic-associated cell division[60]. Notably, the *Streptomyces* chromosome contains an unusually high number of ParB binding sites (16–20), indicating a complex architecture of the segrosome.

Since interarm contacts occur during cell division, a characteristic spatial architecture of sporogenic segrosomes is implied to be a prerequisite for SMC loading. The significance of the ParB complex architecture for SMC loading has been established by

**Fig. 3 Influence of SMC and HupS on chromosome organisation. A** Top panels: the normalised Hi-C contact maps obtained for the wild-type and *smc*, *hupS* and *smc hupS* double mutants (*ftsZ-ypet* derivatives, MD100, TM005, TM004, TM006, respectively) growing for 22 h or for 25 h (only in the case of the double *hupS smc* mutant) (in 5 ml cultures). Bottom panels: the corresponding differential contact maps in the logarithmic scale (log2) comparing the contact enrichment in the wild-type strain (red) versus the mutant strains (blue) are shown below each Hi-C contact map. **B** The number of FLAG-SMC binding sites along the chromosome of the wild-type (blue) (TM017) or *parB* mutant background (red) (KP4F4) determined by ChIP-Seq analysis at the time of sporogenic cell division (14 h of 50 ml culture growth). The number of FLAG-SMC binding sites (250 bp) was determined by *normr* analysis averaged over 0.5 Mbp sliding window every 1000 bp with a loess model fit. Points below the plot show positions of individual sites coloured according to their FDR (false discovery rate) values (yellow less significant, dark purple more significant). **C** The number of HupS-FLAG binding sites along the chromosome in the wild-type background (*hupS* deletion complemented with *hupS-FLAG* delivered in trans, TM015) determined by ChIP-Seq analysis at the time of sporogenic cell division (14 h of 50 ml culture growth). The number of HupS-FLAG binding sites was determined by differential analysis of the HupS-FLAG and wild-type strains using *edgeR* and averaged over 0.5 Mbp sliding window every 1000 bp. Smooth line shows loess model fit. The points below the plot show positions of individual sites coloured according to their FDR values (yellow less significant, dark purple more significant). **D** Heatmaps showing the number of reads for all 307 regions bound by HupS-FLAG in the wild-type background (TM015), *smc* mutant background (TM016) and negative control (the wild-type strain). The reads were normalised by the glmQL model from the *edgeR* package. For each region, position 0 is the position with the maximum number of reads. Regions are sorted according to their logFC values. **E** Comparison of HupS-FLAG binding in the wild-type (blue) and *smc* deletion background (red). The lines show mean values of reads (for 69 bp long regions) with 95% confidence intervals for all 307 sites differentially bound by HupS-FLAG protein. For all sites, 1000 bp long fragments were extracted from the chromosome centred around the best position as determined by *edgeR* analysis.

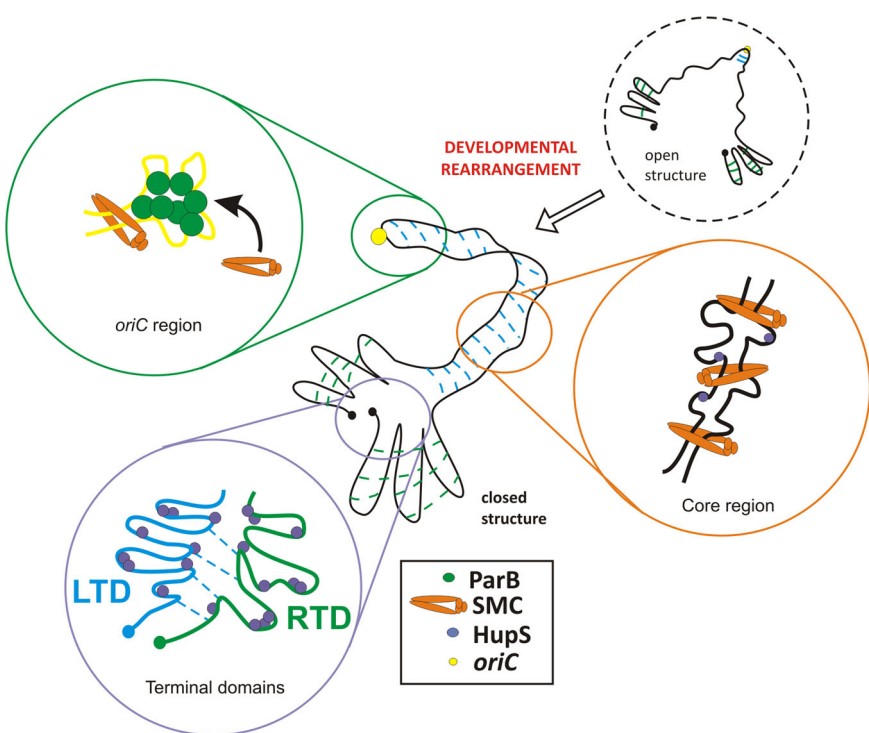

**Fig. 4 Model of spatial rearrangement of the *S. venezuelae* chromosome during sporogenic development showing the contribution of ParB (green), SMC (orange) and HupS (blue) to chromosomal arm alignment and organisation of terminal domains (TLD and RTD).** Blue lines indicated intrachromosomal contacts, *oriC* region is marked with a yellow circle.

studies of ParB mutants with impaired spreading or bridging in *B. subtilis* and in *C. glutamicum*[7,27]. Accordingly, our observations suggest that during *Streptomyces* differentiation ParB complexes spatially rearrange to increase their capability of SMC loading and initiating chromosomal folding with proximal arms.

What factors could induce the reorganisation of ParB complexes during *Streptomyces* sporulation? ParB has recently been shown capable of binding to and hydrolysing CTP, which affects the formation of segrosomes and appears to be conserved among ParB homologues[65–67]. While not yet confirmed in *Streptomyces*, CTP binding and hydrolysis may be involved in the rearrangement of ParB complexes on chromosomes in sporogenic hyphae. The formation of ParB complexes has also been shown to be associated with sporulation-dependent deacetylation of ParB in *S.*

*coelicolor*[68]. Furthermore, segrosome rearrangement may also result from the increased transcriptional activity of the *parAB* operon. ParA accumulates in sporogenic cells, promoting the formation and regular distribution of the segrosomes[39,60]. In other bacteria (*B. subtilis*, *V. cholerae*, and *M. smegmatis*) the elimination of ParA also decreased ParB binding[69–71]. Indeed, our Hi-C mapping showed that in the absence of ParA, interarm interactions are somewhat diminished, reinforcing the role played by ParA in executing ParB architecture. Finally, since the transcriptional activity was proven to have a considerable impact on chromosomal arm alignment[5,63,72], sporulation-associated changes in transcription within the chromosomal core may account for the development of interarm contacts.

Our study has established that although HupS promotes contacts all over the chromosome, it is particularly involved in the organisation of the terminal domains. HupS exhibited binding sequence preference in vivo, but no sequence specificity when binding linear DNA fragments. Thus, we infer that specific DNA structural features may account for enhanced HupS binding in particular regions. Reduced binding of HupS in the absence of SMC could be explained by preferential binding by HupS to structures such as loops, generated by SMC. This notion corroborates earlier suggestions that HU homologues could stabilise DNA plectonemic loops, increasing short-range contacts[6]. *E. coli* studies also suggested cooperation between condensin MukB and HU homologues in maintaining long-range chromosomal contacts[10]. In fact, in *E. coli*, condensin activity was suggested to be increased by DNA structures induced by HU protein[10]. Previous analysis of *S. coelicolor* showed that HupS deletion resulted in chromosome decompaction in sporogenic hyphae[42]. Our Hi-C analyses have established that both SMC and HupS contribute to chromosome compaction during sporulation, and the double *hupS smc* deletion has the strongest effect on chromosomal conformation, affecting both interarm and short-range chromosomal contacts (Fig. 4).

Although elimination of SMC and HupS disturbed the *S. venezuelae* chromosome structure, the phenotype of the double mutant was still rather mild and manifested predominantly in increased nucleoid volume and formation of elongated spores. While a mild phenotype for an *smc* mutant was also described in *C. glutamicum*, in most bacteria (e.g., *B. subtilis. C. crescentus* and *S. aureus*), *smc* deletion results in either chromosome decompaction and/or segregation defects and often led to the formation of elongated cells[7,21,73–75]. Interestingly, condensins have been suggested to be particularly important for chromosome segregation in bacteria lacking the complete *parABS* system (such as *E. coli*)[76–78]. The variation of spore size observed in *S. venezuelae smc* mutants correlates with irregular placement and lowered stability of Z-rings, suggesting that decreased chromosome compaction affects Z-ring formation and/or stability.

In conclusion, we have established the first maps of chromosomal contacts for a linear bacterial chromosome. The *S. venezuelae* chromosome conformation undergoes extensive rearrangement during sporulation. Similar chromosomal rearrangements have been observed during the transition from the vegetative to stationary phase of growth in *S. ambofaciens*[63]. DNA-organising proteins HupS and SMC contribute to chromosome compaction, and they also affect cell division. Although roles for HU and SMC have been established in numerous bacteria genera, in *Streptomyces*, their function is seemingly adjusted to meet the requirements of compacting the large linear chromosomes. Owing to the unique developmental life cycle of *Streptomyces* spp., in which chromosome replication and segregation are spatiotemporally separated, the transition between the particular stages of the life cycle requires chromosomal rearrangement (Fig. 4) on a similar scale and complexity to those seen for chromosomes during the eukaryotic cell cycle.

## Methods

**Bacterial strains and plasmids**. The *S. venezuelae* and *E. coli* strains utilised in the study are listed in Supplementary Tables 1 and 2 respectively, and the plasmids and oligonucleotides are listed in Supplementary Tables 3 and 4, respectively. DNA manipulations were performed by standard protocols[79]. Culture conditions, antibiotic concentrations, transformation and conjugation methods followed the commonly employed procedures for *E. coli*[79] and *Streptomyces*[80].

To construct unmarked *smc* deletion *S. venezueale* strain (TM010), first apramycin resistance cassette was inserted into the chromosome to replace *smc* gene using λ Red recombination based PCR-targeting protocol[81]. First the cassette encompassing *apra* gene and *oriT* was amplified on the template of pIJ773 using primers p_smc_Fw and p_smc_Rv and used to transform BW25113 carrying cosmid Sv-3-B07 and λ RED plasmid, pIJ790. The resulting cosmid Sv-3-B07Δ*smc::apra* was

used for conjugation into wild-type *S. venezuelae*. The Apr^R exconjugants were screened for the loss of Kan^R, indicating a double-crossover allelic exchange of the *smc* locus. The obtained colonies were verified by sequencing of the PCR products obtained using the chromosomal DNA as the template and the primers p_sv_smc_spr_Fw and p_sv_smc_spr_Rv. The obtained strain TM001 was next used for further modifications.

To obtain unmarked *smc* deletion strain, FLP recombinase, that recognises FRT sites flanking *apra* cassette, was used. To this end, DH5α containing pCP20 plasmid, that encodes yeast FLP recombinase, were transformed with the cosmid Sv-3-B07 Δ*smc::apra*. Kan^R and Apr^R resistant transformants were incubated at 42 °C and next clones sensitive to apramycin were selected. After verification of the *scar* sequence generated in *smc* locus, the obtained Sv-3-B07 Δ*smc::scar* cosmid was modified, using PCR-targeting protocol, by insertion of *apra-oriT* cassette in the *bla* locus in the SuperCos. The cassette was amplified at the template of pIJ773 plasmid using primers p_blaP1 and p_blaP2. The resulting cosmid Sv-3-B07 Δ*smc::scar* was used for conjugation into *S. venezuelae smc::apra* (TM001) strain. The Apr^R, Kan^R exconjugants were next screened for the loss of Kan^R Apr^R, indicating a double-crossover allelic exchange of the *smc* locus. The obtained strain TM010 was verified by sequencing of the PCR products obtained using the chromosomal DNA as the template and the primers p_sv_smc_spr_Fw and p_sv_smc_spr_Rv.

Strains producing FLAG-SMC: *FLAG-smc* (TM017) and *parB FLAG-smc* (KP4F4) were constructed by homologous recombination and modification of *smc* gene in its native chromosomal locus. First the apramycin resistance cassette flanked with the oligonucleotide encoding FLAG was amplified on the template of pIJ773, using p_apra_FLAG_Fw and p_apra_FLAG_Rv primers that contained NdeI restriction sites and used to transform BW25113 carrying cosmid Sv-3-B07 and λ RED plasmid, pIJ790. Next, the obtained cosmid Sv-3-B07 *apra-oriT- FLAG-smc* was digested with *Nde*I and religated to remove *apra-oriT* from the upstream region of *smc* gene. The obtained cosmid was verified using PCR with primers p_smc_promoter_Fw and p_smc_NFLAG_Rv and then modified, using PCR-targeting protocol, by insertion of *apra-oriT* cassette in the *bla* locus in the SuperCos. The cassette was amplified at the template of pIJ773 plasmid using primers p_blaP1 and p_blaP2. The resulting disrupted cosmid Sv-3-B07 *FLAG-smc* was used for conjugation into *S. venezuelae smc::apra* (TM001) strain. The Apr^R, Kan^R exconjugates were next screened for the loss of Kan^R Apr^R, indicating a double-crossover allelic exchange of the *smc* locus. The obtained strain TM017 was verified by sequencing of the PCR products obtained with the primers p_smc_promoter_Fw and p_smc_NFLAG_Rv on the template of the chromosomal DNA as well as Western Blotting with the anti-FLAG M2 antibody (Merck).

To obtain *parB FLAG-smc* strain (KP4F4) cosmid Sv-4-A98Δ*parB::apra* was used for conjugation into *S. venezuelae FLAG-smc* (TM017) strain. The Apr^R exconjugants were screened for the loss of Kan^R, indicating a double-crossover allelic exchange of the *parAB* locus.

*S. venezuelae hupS* deletion strains (*hupS* (AKO200), *hupS smc* (TM003) and *hupS FLAG-smc* (TM018)) were constructed using λ Red recombination PCR-targeting method[81] by the introduction of *apra* resistance cassette in the *hupS* locus. First, the cassette encompassing *apra* gene and *oriT* was amplified on the template of pIJ773 using primers p_hupS_Fw and p_hupS_Fw and used to transform BW25113 carrying cosmid Sv-5-D08 and λ RED plasmid, pIJ790. The resulting cosmid Sv-5-D08Δ*hupS::apra* was used for conjugation into the wild-type, *smc::scar* (TM010) and to *FLAG-smc* (TM017) strains. The Apr^R exconjugants were screened for the loss of Kan^R, indicating a double-crossover allelic exchange in the *hupS* locus. The obtained *hupS* (AKO200), *hupS smc* (TM003) and *hupS FLAG-smc* (TM018) strains were verified by sequencing of the PCR products obtained using the chromosomal DNA as the template and the primers p_hupSspr_Fw and p_hupSspr_Rv.

To construct *hupS-FLAG* strains (*hupS*-FLAG (TM015) and *hupS*-FLAG *smc* (TM016)) plasmid pIJ10770-*hupS-FLAG*, that contained *hupS* gene under the control the native *hupS* promoter, was used. The oligonucleotide encoding FLAG was amplified using primers P_pSS170_xhoI_FLAG_Fw and P_pSS170_eco32I_FLAG_Rv that included restriction sites XhoI and Eco32I and cloned into pIJ10770 plasmid (Hyg^R, kind gift from dr. Susan Schlimpert, John Innes Centre, Norwich, UK) delivering pIJ10770-*FLAG*. *hupS* gene with its native promoter was amplified on the template of cosmid Sv-5-D08 using primers P_pSS170_hupS_FLAG_Fw and P_hupS-tocherry-Rv. The fragment containing *hupS* gene was cloned to pIJ10770-*FLAG* using restriction sites Acc65I and XhoI. The obtained plasmid pIJ10770-*hupS-FLAG* was used for conjugation to *hupS* (AKO200) and *hupS smc* (TM003) strains delivering *hupS-FLAG* (TM015) and *hupS-FLAG smc* (TM016). Strains were verified by sequencing of the PCR products obtained using primers p_pSS_spr_Fw and p_pSS_spr_Rv on the template of chromosomal DNA, and western blotting with anti-FLAG M2 antibody (Merck).

To construct the *S. venezuelae* strains expressing *ftsZ-ypet* the obtained strains TM010, AKO200 and TM003 were used for conjugation with the *E. coli* carrying pKF351 *ftsZ-ypet apra* (to TM010) or pKF351 *ftsZ-ypet hyg* (to AKO200 and TM003) plasmid. *smc ftsZ-ypet* (TM004 Apr^R), *hupS ftsZ-ypet* (TM005, Hyg^R) and *hupS smc ftsZ-ypet* (TM006, Hyg^R) were verified by PCR with p_ftszypet_fw and p_ftszypet_Rv and fluorescence microscopy.

To construct the *S. venezuelae* strains expressing *ftsZ-ypet* and *hupA-mcherry*, we used plasmid pSS172hupA-mcherry (hyg^R, kind gift from dr. Susan Schlimpert, John Innes Centre, Norwich, UK) that contained *hupA-mCherry* under control of the native *hupA* promoter. Plasmid pSS172*hupA-mCherry* was introduced to *ftsZ-ypet* (in the wild-type background, MD100, Apr^R) and *smc ftsZ-ypet* (TM004,

Apr[R]) strain generating *ftsZ-ypet hupA-mcherry* (TM011, Apr[R], Hyg[R]) and *smc ftsZ-ypet hupA-mCherry* (TM012, Apr[R], Hyg[R]), respectively. Strains *hupS ftsZ-ypet, hupA-mCherry* (TM013, Apr[R], Hyg[R] Spec[R]) and *hupS smc ftsZ-ypet hupA-mCherry* (TM014, Apr[R], Hyg[R] Spec[R]) were constructed by subsequent delivery of pKF351 *ftsZ-ypet spec* and pSS172*hupA-mCherry* to *hupS* (AKO200) and *hupS smc* (TM003). The obtained strains were verified by PCR with P_{pSS_spr_Fw} and P_{pSS_spr_Rv} primers using chromosomal DNA as template and by using fluorescence microscopy.

Genetic complementation of Δ*smc::scar* (TM010) was performed by delivering smc-FLAG in trans. To this end the *smc* gene was amplified using cosmid DNA Sv-3B07 as template, using primers p_{smc_poczatek_Fw} and p_{smc_exp_Rv}, next PCR product was cloned into pGEM-T-Easy vector, to which double-stranded oligonucleotide encoding FLAG was inserted using BamHI and Mph1103I restriction enzymes. The promoter region of *smc* was amplified using Sv-3-B07 cosmid DNA as template and primers p_{smc_promoter_Fw} and p_{smc_promoter_Rv} and cloned directly into pMS83 using KpnI and NdeI. Next, fragment encoding SMC-FLAG was cloned from pGEM-TEasy-*smc-FLAG* to pMS83-*p_{smc}* yielding pMS83-*smc-FLAG*, which was used for conjugation to Δ*smc::scar* (TM010) yielding TM019 (Δ*smc::scar*; pMS83-*smc-FLAG*). The construct was verified by sequencing of PCR products obtained using the chromosomal TM019 as template and primers p_{smc_spr_Fw} and p_{smc_spr_Rv;} as well as p_{smc_inter_Fw} and p_{smc_NFLAG_Rv,.} and then by western blotting.

## *S. venezuelae* cultures and growth rate analysis

To obtain the standard growth curves comparing the growth rate of *S. venezuelae* strains, a Bioscreen C instrument (Growth Curves US) was used. Cultures (at least three independent cultures for the same strain) in liquid MYM medium (300 µl) were set up by inoculation with $6 \times 10^3$ colony-forming units (CFUs) of *S. venezuelae* spores. The cultures grew for 48 h at 30 °C under the 'medium' speed and shaking amplitude settings, and their growth was monitored by optical density measurement ($OD_{600}$) every 20 min. The data were collected using BioScreener 3.0.0 software.

The Hi-C cultures were set up as follows: 5 ml of MYM medium was inoculated with $10^8$ CFUs, and the cultures were incubated for 6–26 h at 30 °C with shaking (180 rpm). For the '5 ml culture' growth curve depiction, the optical density ($OD_{600}$) of the culture diluted (1:10) with MYM medium was measured in 1-h intervals.

## Preparation of Hi-C libraries and data analysis

For the preparation of *S. venezuelae* Hi-C contact maps, 5 ml cultures were established as described above. Each *S. venezuelae* strain and/or developmental stage was analysed in at least 2 experimental replicates. The cultures were incubated for 13 to 25 h at 30 °C with shaking (180 rpm). After this step, the cultures were cross-linked with 1% formaldehyde (Sigma Aldrich) for 7 min and 30 s and blocked with 0.2 M glycine for 15 min, both steps performed at 30 °C with shaking (180 rpm). One millilitre of the cross-linked culture was centrifuged (1 min, 2400 g at room temperature). The mycelium pellet was washed twice with 1.2 ml of TE buffer (10 mM Tris-HCl pH = 8.0, 1 mM EDTA) and resuspended in 0.6 ml of TE buffer. Two vials, each containing 25 µl of the resuspended mycelium, were independently processed according to the protocol below (a modification of the protocol described by Le et al.[82]). The cells were lysed using 0.5 µl of ReadyLyse solution (Invitrogen) at 37 °C for 30 min followed by the addition of 1.25 µl of 5% SDS solution (Sigma Aldrich) and incubated at room temperature for 15 min. After cell lysis, the reaction was supplemented with 5 µl of 10% Triton X-100 (Sigma Aldrich), 5 µl of 10× reaction buffer 3 (New England Biolabs) and 10.75 µl of DNase-free water (Invitrogen) and subsequently incubated for 15 min at room temperature. The chromosomal DNA was digested with 2.25 µl of *BglII* restriction enzyme (New England Biolabs; 50,000 U/ml) for 2 h and 30 min at 37 °C followed by the addition of 0.25 µl of *BglII* (New England Biolabs; 50,000 U/ml) and incubation for another 30 min at 37 °C. 50 µl of the *BglII*-digested reaction was cooled down on ice and next, the generated sticky ends were labelled with biotin-14-dATP (Invitrogen) by addition of 0.9 µl of each: 2 mM dGTP, dTTP and dCTP (New England Biolabs) as well as 4.5 µl of 0.4 mM biotin-14-dATP (Invitrogen), 1.6 µl of ultrapure water and 1.2 µl of Klenow large fragment (New England Biolabs; 5000 U/ml). The reaction mixtures were incubated for 45 min at room temperature and subsequently stopped with 3 µl of 5% SDS (Sigma Aldrich). In the next step, the filled-in DNA ends were ligated by addition to the reaction mixture: 75 µl of 10% Triton X-100, 100 µl of 10× T4 ligation buffer (New England Biolabs), 2.5 µl of bovine serum albumin (New England Biolabs; 10 mg/ml) and 800 µl ultrapure water. The reaction mixtures were incubated for 15 min on ice, subsequently supplemented with 3 µl of T4 DNA ligase (New England Biolabs, 2,000,000 U/ml), and incubated overnight in ice-bath. Next day, the ligation reaction was stopped by the addition of 20 µl of 0.5 M EDTA pH = 8.0 (Invitrogen) and 2.5 µl of proteinase K (New England Biolabs, 20 mg/ml) followed by incubation at 65 °C for 6–8 h. Next, the samples were extracted twice with one volume of phenol/chloroform/isoamyl alcohol (25:24:1) pH = 8.0 (Sigma Aldrich). The water phase containing DNA was next mixed with one volume of isopropanol (Sigma Aldrich) supplemented with 3 µl of GlycoBlue co-precipitant (ThermoFisher Scientific) and incubated overnight at −80 °C. Next, two parallelly processed samples containing precipitated DNA were combined and resuspended in ultrapure water in final volume 60 µl. Next, the DNA was sonicated to reach DNA in average size 200–500 bp, resolved in 1% agarose gel and purified using Qiaquick Gel Extraction Kit (Qiagen). The shared DNA was

eluted with 50 µl of ultrapure water. To repair DNA ends after fragmentation, 50 µl of the shared DNA from the previous step was supplemented with 10 µl of 10× T4 DNA ligase buffer (New England Biolabs), 2.5 µl of 10 mM dNTPs (New England Biolabs), 28.75 µl of ultrapure water, 4 µl of T4 DNA polymerase (New England Biolabs, 3000 U/ml), 4 µl of T4 polynucleotide kinase (New England Biolabs, 1000 U/ml) and 0.75 µl of Klenow large fragment (New England Biolabs, 5000 U/ml), and followed by incubation at room temperature for 30 min. After the reaction, DNA was purified using MinElute Reaction CleanUp Kit (Qiagen). From the column, DNA was eluted with 30 µl of ultrapure water. In the next step, 3'-adenine-overhangs were added to the repaired DNA by incubation of 30 µl of the reaction from the step above for 45 min at room temperature with 4 µl of 10× reaction buffer 2 (New England Biolabs), 4 µl of 2 mM dATP (New England Biolabs) and 3 µl of Klenow 3'–5' exo⁻ (New England Biolabs, 5000 U/ml). Next, DNA was purified using MinElute Reaction CleanUp Kit (Qiagen) and eluted with 15 µl of ultrapure water. The NEBNext adaptors (5 µl) (New England Biolabs) were subsequently ligated according to the manufacture's protocol. In the next step, the biotin-labelled DNA was purified from non-labelled DNA using DynaBead MyOne Streptavidin C1 magnetic beads (Invitrogen). First, 25 µl of magnetic beads were washed twice with 200 µl of NTB buffer (5 mM Tris-HCl pH 8.0, 0.5 mM EDTA, 1 M NaCl) and twice with 200 µl of ultrapure water. Finally, the magnetic beads were resuspended in 10 µl of ultrapure water and transferred to the NEBNext adaptors ligation mixture (from the earlier step). The pulled-down beads containing biotinylated DNA were washed twice with 200 µl of NBT buffer, twice with 200 µl of ultrapure water and subsequently resuspended with 12 µl of ultrapure water 1.2 µl of beads resuspension was used immediately as a template for Hi-C library preparation as well as NEBNext Index and NEBNext Universal PCR primers (New England Biolabs). The DNA was amplified using Phusion DNA polymerase enzyme (ThermoFisher Scientific) according to the manufacturer's protocol. The number of PCR amplification cycles was limited to 18. The amplified Hi-C libraries were purified from a 1% agarose gel, and their concentrations were measured using a Qubit dsDNA HS Assay Kit (ThermoFisher Scientific) in OptiPlate-96 White (Perkin Elmer) with 490-nm and 520-nm excitation and emission wavelengths, respectively. The agarose-purified Hi-C libraries were diluted to a 10 nM concentration, pooled together, and sequenced using Illumina NextSeq500 ($2 \times 75$ bp; paired reads), as offered by the Fasteris SA (Plan-les-Ouates, Switzerland) sequencing facility. Sequencing data were collected and pre-processed by Fasteris SA using the software as below: NextSeq Control Software 4.0.1.41, RTA 2.11.3 and bcl2fastq2.17 v2.17.1.14.

Complete data processing was performed using the Galaxy HiCExplorer web server[83]. The paired reads were first trimmed using *Trimmomatic* (version 0.36.5) and subsequently filtered using *PRINSEQ* (version 0.20.4) to remove reads with more than 90% GC content. The pre-processed reads were mapped independently to the *S. venezuelae* chromosome using *Bowtie2* (version 2.3.4.3; with the *-sensitive -local* presets) (Supplementary Fig. 1A). The Hi-C matrix was generated using *hicBuildMatrix* (version 2.1.4.0) with a bin size of 10,000 bp. In the next step, three neighbouring bins were merged using *hicMergeMatrixBins* (version 3.3.1.0) and corrected using the *hicCorrectMatrix* tool (version 3.3.1.0) based on Imakaev's iterative correction algorithm[84] (with the presets as follows: number of iterations: 500; skip diagonal counts: *true* (a standard pipeline) to highlight secondary diagonal contacts, or *false* to highlight primary diagonal contacts (used only for wild-type 22 h; presented in Fig. 1B); remove bins of low coverage: −1.5; remove bins of high coverage: 5.0). The threshold values for the low and high data coverage were adjusted based on the diagnostic plot generated by the *hicCorrectMatrix* tool (version 3.3.1.0). The corrected Hi-C matrixes were normalised and compared using *hicCompareMatrixes* (version 3.4.3.0, presets as below: *log2ratio*), yielding differential Hi-C contact maps. The differential maps were plotted using *hicPlotMatrix* (version 3.4.3.0; with masked bin removal and *seismic* pallets of colours). To prepare the Hi-C contact map, the corrected matrixes were normalised in the 0 to 1 range using *hicNormalize* (version 3.4.3.0; settings: set values below the threshold to zero: value 0) and subsequently plotted using *hicPlotMatrix* (version 3.4.3.0; with the removal of the masked bins), yielding 30-kbp-resolution Hi-C heatmaps. The threshold values (min/max) were manually adjusted for each Hi-C contact map independently, and the min and max values oscillated in the ranges of 0.05–0.1 and 0.3–0.4, respectively. The corrected matrixes served as the input files for the multiple pairwise matrix comparison using *hicCorrelate* (version 3.6 + galaxy0). The Hi-C contact matrixes were compared and grouped based on the calculated Pearson correlation coefficients (Supplementary Fig. 1B). To identify the LTD and RTD domain organisation, principal component analysis (PCA1) was performed using *hicPCA* (version 3.6 + galaxy0). The generated bedgraph files were extracted and smoothed using the moving average, and the PCA1 scores for each 30 kbp bin were plotted against the *S. venezuelae* chromosome position. The domain boundaries were identified based on the position of the *X* axis intersection.

## Chromatin immunoprecipitation combined with next-generation sequencing (ChIP-seq)

For chromatin immunoprecipitation, *S. venezuelae* cultures (three independent cultures for each analysed strain) were grown in 50 ml liquid MYM medium supplemented with Trace Element Solution (TES, 0.2× culture volume)[80] at 30 °C with shaking (ChIP-seq cultures). Cultures were inoculated with $7 \times 10^8$ CFUs. After 14 h of growth, the cultures were cross-linked with 1% formaldehyde for 30 min and blocked with 125 mM glycine. Next, the cultures were washed twice

with PBS buffer, and the pellet obtained from half of the culture volume was resuspended in 750 µl of lysis buffer (10 mM Tris-HCl, pH 8.0, 50 mM NaCl, 14 mg/ml lysozyme, protease inhibitor (Pierce)) and incubated at 37 °C for 1 h. Next, 200 µl of zircon beads (0.1 mm, BioSpec products) were added to the samples, which were further disrupted using a FastPrep-24 Classic Instrument (MP Biomedicals; 2 × 45 s cycles at 6 m/s speed with 5-min breaks, during which the lysates were incubated on ice). Next, 750 µl IP buffer (50 mM Tris-HCl pH 8.0, 250 mM NaCl, 0.8% Triton X-100, protease inhibitor (Pierce)) was added, and the samples were sonicated to shear DNA into fragments ranging from 300 to 500 bp (verified by agarose gel electrophoresis). The samples were centrifuged, 25 µl of the supernatant was stored to be used as control samples (input), and the remaining supernatant was used for chromatin immunoprecipitation and mixed with magnetic beads (60 µl) coated with anti-FLAG M2 antibody (Merck). Immunoprecipitation was performed overnight at 4 °C. Next, the magnetic beads were washed twice with IP buffer, once with IP2 buffer (50 mM Tris-HCl pH 8.0, 500 mM NaCl, 0.8% Triton X-100, protease inhibitors (Pierce)), and once with TE buffer (Tris-HCl, pH 7.6, 10 mM EDTA 10 mM). DNA was released by overnight incubation of beads resuspended in IP elution buffer (50 mM Tris-HCl pH 7.6, 10 mM EDTA, 1% SDS) at 65 °C. The IP elution buffer was also added to the 'input' samples, which were treated further as immunoprecipitated samples. Next, the samples were centrifuged, and proteinase K (Roche) was added to the supernatants to a final concentration of 100 µg/ml followed by incubation for 90 min at 55 °C. DNA was extracted with phenol and chloroform and subsequently precipitated overnight with ethanol. The precipitated DNA was dissolved in nuclease-free water (10 µl). The concentration of the DNA was quantified using a Qubit dsDNA HS Assay Kit (ThermoFisher Scientific). DNA sequencing was performed by Fasteris SA (Switzerland) using the Illumina ChIP-Seq TruSeq protocol, which included quality control, library preparation and sequencing from both ends ($2 \times 75$ bp) of amplified fragments. Sequencing data for HupS-FLAG ChIP-Seq experiment were collected and pre-processed by Fasteris SA using software: HiSeq Control Software 4.0.1.41, RTA 2.7.7, bcl2fastq2.17v2.17.1.14. Sequencing data for FLAG-SMC ChIP-Seq experiment were collected and pre-processed by Fasteris SA using software: NovaSeq Control Software 1.6.0, RTA 3.4.4, bcl2fastq2.20 v2.20.0.422.

The complete bioinformatic analysis was performed using the R packages *edgeR* (3.30.3), *normr* (1.14.0), *rGADEM* (2.36) and *csaw* (1.22.1), as well as the *MACS2* (2.2.7.1) and *MEME* (5.1.0) programmes. HupS-FLAG binding was analysed using an earlier published protocol[85]. The mapping of ChIP-seq data was performed using the *Bowtie2* tool (version 2.3.5.1[86,87]). The successfully mapped reads were subsequently sorted using *samtools* (version 1.10)[88]. The total number of mapped reads was above $10^6$ on average. Regions differentially bound by HupS-FLAG were identified using the R packages *csaw* and *edgeR*[89–92], which also normalised the data, as described earlier, as follows[85]. The mapped reads were counted within a 69-bp-long sliding window with a 23-bp slide. Peaks were filtered using the local method from the *edgeR* package, which compares the number of reads in the region to the background computed in the surrounding window of 2000 bp. Only regions with log2-fold change above 2.0 were utilised for further analysis. Each window was tested using the QL F-test. Regions that were less than 100 bp apart were merged, and the combined *p*-value for each was calculated. Only regions with false discovery rate (FDR) values below the 0.05 threshold were considered to be differentially bound. The identified regions were further confirmed by analysis with the *MASC2* programme using the *broad* option[93]. The R packages *rGADEM* and *MEME* Suite were used to search for probable HupS binding sites[94]. FLAG-SMC bound regions were analysed using the R package *normr* using Input ChIP-seq data as a control. Reads were counted in 250-bp-long regions, and a region was considered to be bound by FLAG-SMCs if the FDR value compared to the Input value was below the 0.05 threshold and if it was not found to be significant in a control strain not producing FLAG-SMC protein[95].

**Quantification of the *oriC/arm* ratio.** To estimate the *oriC/arm* ratio, chromosomal DNA was extracted from *S. venezuelae* 5 ml cultures (as for Hi-C) growing for 13–26 h or from 50 ml culture (as for ChIP-seq) growing for 8 to 16 h. One millilitre of each culture was centrifuged (1 min at 2400 g and at room temperature). The supernatant was discarded, and the mycelium was used for the subsequent isolation of chromosomal DNA with application a Genomic Mini AX *Streptomyces* kit (A&A Biotechnology) according to the manufacturer's protocol. The purified DNA was dissolved in 50 µl of DNase-free water (Invitrogen). The DNA concentration was measured at 260 nm and subsequently diluted to a final concentration of 1 ng/µl.

qPCR was performed using Power Up SYBR Green Master Mix (Applied Biosystems) with 2 ng of chromosomal DNA serving as a template for the reaction and oligonucleotides complemented to the *oriC* region (*oriC*) (gyr2_Fd, gyr2_Rv) or to the right arm termini (*arm*) (arg3_Fd, arg3_Rv) (Supplementary Table 4). The changes in the *oriC/arm* ratio were calculated using the comparative ΔΔCt method, with the arm region being set as the endogenous control. The *oriC/arm* ratio was estimated as 1 in the culture (5 ml) growing for 26 h, corresponding to the appearance of spore chains.

**Light microscopy.** The analyses of spore sizes were performed using brightfield microscopy. Spores from 72-h plate cultures (MYM medium without antibiotics) were transferred to microscopy slides. The samples were observed using a Zeiss Microscope (Axio Imager M1), and images were analysed using Fiji software (ImageJ 2.0.0 or ImageJ 1.53c).

For the visualisation of Z-rings, strains expressing *ftsZ-ypet* as the second copy of the *ftsZ* gene were used (the modification did not affect the growth rate, Supplementary Fig. 2A). For the snapshot analysis of *S. venezuelae* development in liquid, 5 µl from the 5 ml culture (as for Hi-C) growing for 18–26 h or from the 50-ml culture (as for ChIP-Seq) growing for 8–16 h was spread on a coverslip and fixed by washing twice with absolute methanol. For the snapshot analysis of nucleoid areas, strains were cultured on coverslips inserted in MM agar plates supplemented with 1% mannitol[80], and after 22–24 h of growth, they were fixed with 2.8% formaldehyde in PBS. The nucleoids were subsequently stained for 5 min at room temperature with 7-amino-actinomycin D (1 mg/ml 7-AAD in DMSO, ThermoFisher Scientific) diluted 1:400 in PBS buffer. In the next step, the mycelium was washed twice with PBS buffer, and the coverslip was mounted using 50% glycerol in PBS buffer. Microscopic observation of FtsZ-YPET and DNA stained with 7-AAD was performed using a Zeiss Microscope (Axio Imager M1). The images were analysed with dedicated software (Axio Vision Rel software). To calculate the nucleoid separation, the distance between the two neighbouring DNA-free areas was measured for 250 nucleoids. The boxplots showing the length of nucleoid-covered areas were constructed using the Plotly tool (plotly.com).

Time-lapse fluorescence microscopy was performed using CellAsic Onix system according to a previously described protocol[55] as follows: B04A microfluidic plates (ONIX; CellASIC) were washed with MYM medium, next 100 µl of spores were loaded into microfluidic plate at pressure 4 psi repeatedly for 2–10 s, dependently on spore suspension density. Cultures were started by perfusing the spores with MYM medium at constant flow rate (at pressure 3 psi and temperature 30 °C). For growth rate analysis, the extension of the hyphae emerging from spores was measured within 1 h. For sporulation analyses, after 3 h of preincubation, spent MYM, derived from 40 h flask culture and filter-sterilised, was applied to flow channel. The images were taken every 10 min for ~20 h. The DeltaVision Elite Ultra High-Resolution microscope equipped with a DV Elite CoolSnap camera. Resolve3D softWoRX-Acquire Version:6.1.1 software, and ×100, 1.46 NA, DIC lens was used. Samples were exposed for 50 ms at 5% transmission in the DIC channel and for 80 ms at 50% transmission in the YFP and mCherry channels. The time-lapse movies were analysed using ImageJ Fiji software. For nucleoid visualisation in time-lapse microscopy, strains expressing *hupA-mCherry* as an additional copy of the *hupA* gene were used. Nucleoid area was measured using HupA-mCherry as the marker at the time of disassembly of Z-rings (visualised using FtsZ-Ypet fusion), which was ~120 min after their appearance and hyphal growth cessation. Analysis of the HupA-mCherry-labelled chromosome images was based on determining the area and Feret diameter (the maximal distance between two points) of nucleoid fluorescence using the Versatile Wand Tool with 8-connected mode and manually adjusted value tolerance. The distances between Z-rings were measured 60 min after hyphal growth cessation for ~300 Z-rings. The fluorescence intensity of the Z-rings was collected along and averaged across the hyphae with a line selection tool whose width was adjusted to the width of the measured hyphae. The fluorescence intensity of each Z-ring in 25–30 hyphae of each strain was subsequently determined by analysing the data with RStudio using a custom shiny application *findpeaks* based on the *Peaks* R package. Tracking of Z-rings in the time-lapse images was performed using the nearest neighbour algorithm. Only Z-rings observed for more than 90 min were used in the analysis. The data from each time point were combined to show how the standard deviation of fluorescence intensity changed during hyphal development. Representative hyphae of each strain were selected, the FtsZ-YPet fluorescence intensity and Z-ring position were determined using the script described above, and its variability was visualised in the form of a kymograph with a custom script based on the *ggplot2* package. Statistical analysis was performed using the R programme, for data preparation dplyr 1.0.2 was used. Since the variances in the analysed samples were not homogenous, one-way ANOVA with a two-sided Games–Howell post-hoc test was used to assess the differences between sample means. This test compares the difference between each pair of means with an appropriate adjustment for multiple testing.

**Scanning electron microscopy.** *Streptomyces* samples were mounted on an aluminium stub using Tissue Tek^R OCT (optimal cutting temperature) compound (BDH Laboratory Supplies, Poole, England). The stub was then immediately plunged into liquid nitrogen slush at approximately −210 °C to cryo-preserve the material. The sample was transferred onto the cryo-stage of an ALTO 2500 cryo-transfer system (Gatan, Oxford, UK) attached to an FEI Nova NanoSEM 450 (FEI, Eindhoven, The Netherlands). Sublimation of surface frost was performed at −95 °C for ~3 min before sputter coating the sample with platinum for 150 s at 10 mA, at colder than −110 °C. After sputter-coating, the sample was moved onto the cryo-stage in the main chamber of the microscope, held at −125 °C. The sample was imaged at 3 kV and digital TIFF files were stored.

**Reporting summary**. Further information on research design is available in the Nature Research Reporting Summary linked to this article.

## Data availability

The Hi-C data generated in this study (shown in Figs. 1B–D, 3A and Supplementary Figs. 6A–C, 10C) have been deposited in the ArrayExpress database (EMBL-EBI) under accession code E-MTAB-9810. The raw ChIP-Seq data, as well as the processed data generated in this study (shown in Fig. 3C), have been deposited in the ArrayExpress database (EMBL-EBI) under accession code E-MTAB-9821. The raw ChIP-Seq data, as well as the processed data generated in this study (shown in Fig. 3B), have been deposited in the ArrayExpress database (EMBL-EBI) under accession code E-MTAB-9822. The microscopy data are available in Figshare as follows: data shown in Fig. 2A: [https://doi.org/10.6084/m9.figshare.15042900], the microscopy images used for Fig. 2B: [https://doi.org/10.6084/m9.figshare.15043329], and for Supplementary Fig. 8B. [https://doi.org/10.6084/m9.figshare.15043359], the time-lapse movies used for Fig. 2E, F and Supplementary Fig. 7D: [https://doi.org/10.6084/m9.figshare.15042801] and for Supplementary Fig. 7A, inset: [https://doi.org/10.6084/m9.figshare.15043338]. The images used for Supplementary Fig. 4B: [https://doi.org/10.6084/m9.figshare.15042921]. Source data are provided with this paper.

## Code availability

The code used to determine FtsZ foci positions in time-lapse fluorescence profiles: https://github.com/astrzalka/findpeaks.

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

## Acknowledgements

This work was funded by the Polish National Science Centre: HARMONIA grant 2016/22/NZ1/00122 (to M.J.S.) and OPUS grant 2018/31/B/NZ1/00614 (to D.J.). We are grateful to Susan Schlimpert for sharing pIJ10770 and pSS172. We thank the Bioimaging facility of the John Innes Centre, supported by a core capability grant from BBSRC, for use of their resources. We thank Jolanta Zakrzewska-Czerwińska for discussions and comments on the manuscript. T.B.K.L. acknowledges the Royal Society University Research Fellowship (UF140053) and BBSRC (BBS/E/J/000PR9791) for funding.

## Author contributions

Conceptualisation and experiment design (M.J.S., D.J.). Methodology including strains construction (T.M., A.Z., A.K.-O., K.P.), epifluorescence microscopy observation and data analysis (T.M., M.J.S., K.P., A.S.); electron microscopy observations (K.F.); introduction to the Hi-C technique (T.B.K.L.); Hi-C libraries preparation and data analysis (M.J.S.); ChIP-Seq libraries preparation and data analysis (T.M., K.P., A.S.); western blots and in vitro DNA binding assays (T.M., K.P., J.D.); growth curves (M.J.S.); ori/arm ratio analyses (M.J.S.). Figures preparation and writing the manuscript (M.J.S., D.J., A.S.); manuscript revision and discussion (T.B.K.L., A.S.).

## Competing interests

The authors declare no competing interests.
