## [Peer Review File · Nature Communications]

REVIEWER COMMENTS

Reviewer #1 (Remarks to the Author):

The manuscript by Szafran et al. is a very nice study on the structure of the linear *Streptomyces venezuelae* chromosome as the filamentous bacterium leaves vegetative growth, enters its developmental cycle and completes spore formation. The information is novel and the experiments have been carefully performed and well presented.

1) As a reader, the reviewer had difficulty really seeing the diffuse signal suggesting spatial proximity (lines 336-338) and contacts at the termini in Fig. 1B. The manuscript would be strengthened by spending a little more effort presenting this in the figure. Where is the diffuse signal? One needs to spend time looking at those types of plots and many people have not had much experience. What parts of the data in Figure 1 indicates that LTD and RTD are "organized"? This becomes one of the main conclusions of the study and would benefit from helping the reader early on in the manuscript.

2) The structure of the *Streptomyces* chromosome is unknown to most other microbiologists that naturally think of a circular molecule with all unique coding sequences. Consider adding more of a description on the structure of linear chromosome in the introduction (either after line 72 or line 93). Information might be included is the fact that they have Terminal Inverted Repeats (TIR), and what the length of TIR is for *S. venezuelae*. That there are proteins covalently bound ends that are involved in replication of the ends of the lagging strand ("the end problem") and are believed to interact for the ends of the linear chromosome to colocalize (line 94 and line 504). That there might be intermolecular end-to-end contacts before septation, in addition to intramolecular contacts as shown in the model of Figure 4.

3) Continuing on the fact that the *Streptomyces* chromosome is linear and in reference to "Quantification of the oriC/ter ratio" (line 221), results line 304 and the legend to Figure 1. In *E. coli*, ter is a specific cis-acting DNA sequence. The ter sequences are specific locations opposite oriC to stall and dismantle replication forks. To the reviewer's knowledge, the *Streptomyces* chromosome lacks ter sequences. The work in the manuscript uses the unique argG gene located more than 1 Mb from one end of the chromosome to determine when replication occurs. That should be more clearly and more correctly reflected in the manuscript. Similar issues are located in Figure S6A and legend.

4) Also continuing on the fact that the *Streptomyces* chromosome is linear, what is being shown with the cartoon of the chromosome in the inset of Figure 1A for the measurement of replication. It looks more like theta replication of a circular molecule and less like replication of a linear molecule. Most people would interpret the black dots on the circles as oriC (?) and not ends of chromosomes held together by terminal proteins (?).

5) The manuscript states that the SMC-FLAG protein is stable and the level is constant throughout the growth cycle (lines 438-439), but the stability of the FLAG-SMC seems to be affected in Figure S9C. The band at 12 h is larger than all of the other time points and there is a shadow of a band for full length in the 14-20 h time points. Should there be some concern about the tagged protein that actually accumulates in the cell and used for the ChIP-seq experiments?

Comments about Methods:

6) Line 132 – temperature of crosslinking?

7) Line 140 – 10X or 1X NEB buffer?

8) Line 152 – Location of the sequencing facility?

9) Line 172 – Concentration and pH of the TES supplement?

10) The Principle Component Analysis is not in the methods, but introduced in line 327 and Figure

S2A. In the main text on line 327, only the abbreviation is used (PCA1/PCA2). The abbreviation should be spelled out at the first use.

Cosmetic comments:

11) Line 21 – consider adding the word “central” to read “aligned within the central core region”.

12) Line 24 – For the broad audience, add “HU-family” to read “Streptomyces-specific HU-family protein HupS”. The designation HU is more widely known than HupS. Consider a similar addition to line 389 as well.

13) Lines 35-36 contain some inadvertent duplicated text.

14) Lines 63-64 refer to work in 3 organisms but all three references are for one organism (*B. subtilis*).

15) Line 84 references – The best reference for Z-ring spacing in sporogenic hyphae would be the original Schwedock et al. paper, or other *Streptomyces* FtsZ papers rather than the one on DynAB.

16) Line 309 – The extent of time of this pre-sporulation phase is probably 6-7 h instead of the listed 8 h? Using the manuscript time ranges listed in lines 305 and 309: 19-13=6 h and 21-14=7 h.

17) Line 324- Consider adding the growth time point (22h) in the sentence of the main text.

18) Line 349 – Consider adding the time 25h after “the final stage of spore maturation (25h), to be complete and consistent with the other times pointed out in the lines just preceding that.

19) Line 350 – Consider referring specifically to S1B rather than S1 to read “(Fig. 1A and S1B)”.

20) Line 359 – The correct Figure to reference is Fig. S1B for nucleoid compaction.

21) Lines 376, 378, 381 and 383 – The reference should be to Fig 1D instead of Fig 1C.

22) Line 401 – Beginning here and at other places throughout, hupS-FLAG is correct, but the construct is FLAG-smc. Half of the times the manuscript used FLAG-smc and the other half smc-FLAG, including figures (e.g. Fig. 1B and S7B).

23) Line 409 – The manuscript text should read “Z-ring spacing”. The cells would need to be stained for cell walls to be able to state “pre-spore compartments”.

24) Line 429 – Should Fig. 1D be listed and not Fig. 1B?

25) Line 438 – Spelling of the word “remained”.

26) Line 475 – For clarity, consider changing “the low level” to either “a low level” or “an altered level” of HupS-FLAG.

27) Figure 1A – the scale bar in the microscopy image would be easier to see if it were wider and white on the black background. It is very difficult to see as the current thin red line. Are there white dashed lines on the Hi-C map in Figure 1B? It looks like there might be, but nothing is stated in the legend.

28) Figure 1A legend – Line 829 stated that the oriC/ter ratio for 25 h was set as 1.0, but the plotted value is higher than 1.0.

29) Legend to Figure 2 – hup-mCherry and mCherry-hup are used back and forth in lines 859, 860, 864 and 865. The correct designation should be used consistently.

- 30) Legend to Figure S2 – Refer to Figure 1B in line 2 with the 22h data?
- 31) Legend to Figure S3 – Indicate growth was monitored at A600 for the Bioscreen C instrument? Are the 3 biological replicates all independent from the data in Figure 1D? or is one a repeat of the data in Figure 1D?
- 32) Figure S4D – The y axis label should be “Z-ring spacing” not “compartment length”.
- 33) Figure S6 legend states that the ratio at 16h was set at 1.0, but the plot show that it is about 1.8.
- 34) Figure S7C legend – Are the forward and reverse strands relative to the direction of replication?
- 35) Table S3 – Should “oripUC” be listed as a characteristic of pIJ773 and “meijsca” as a characteristic of pIJ10700?
- 36) Table S4 – There is a mistake and primers Pgyr2_Rv and Pgarg3_Fd have the same sequence listed. It also appears that primer Psmc_promoter_Rv is missing from the table.
- 37) Supplementary information, line 1 under Construction of smc mutant strain – apramycin is misspelled.

Reviewer #2 (Remarks to the Author):

Bacteria organize their chromosomes in a compacted structure, called nucleoid. This organization is essential to fit the large dimensions of the DNA strand into the tiny volume of the cell. Simultaneously, the compacted chromosome need to be replicated and accessible for transcription. Thus, the dynamic arrangement is organized in bacteria by a set of conserved proteins. Several nucleoid associated factors contribute to folding. In particular bacterial condensins (SMC) are required for loop formation and often for replichore cohesion. SMCs are loaded onto the chromosome by DNA-binding proteins, termed ParB. ParB binds to specific DNA sequences termed parS sites. These sites are usually scattered around the origin of replication. *Streptomyces* species display a fascinating life cycle. Vegetative hyphae contain multiple chromosomes and do not require regular cell division. Stationary cells develop into exospores. Spore formation requires coordinated DNA replication and cell division events, combined with precise chromosome segregation. Chromosome folding has been studied only in few model species.

The authors of this manuscript analyzed the chromosomal organization of the Actinobacterium *Streptomyces venezuelae*. Like other *Streptomyces* this strain contains a linear chromosome and thus differs from most other bacteria. So far a detailed, HiC based analysis has not been performed in these bacteria. Therefore, this work is a valuable addition to the existing literature and adds to the completion of our understanding of bacterial chromosomal organization. The main findings are that the *Streptomyces* chromosome are organized by ParB-SMC action. In contrast to other bacteria, there is little arm cohesion during vegetative growth. However, during onset of sporulation the two chromosome arms get zipped up by SMC. SMC loading is dependent on ParB, similar to the situation of other bacteria. The terminal regions of the linear chromosome are folded into distinct domains, by action of the nucleoid associated protein HubS.

This paper contains an important set of data and is well written. The experiments are conducted with great care. Overall, this is a beautiful study. I have only a few questions that should be addressed in a revision.

I interpret the strain list that all experiments were done with a strain carrying the *ftsZ*-ypet allele as sole copy of *ftsZ*. How does this strain grow in comparison to wild type? Earlier publications describe an effect on sporulation (delay). This might be discussed.

It would be helpful if the authors can show where the parS sites are distributed in the *S. ven.* chromosome. In Fig. 1B the dotted line indicates the origin of replications. Does this mean the *Streptomyces* chromosome is organized from the oriC region? This would be an interesting

difference, since other bacterial chromosomes are organized from their parS region. This aspect might be discussed.

Figure 1: The marker frequency data are difficult to see in this arrangement. Consider restructuring. The differential maps based on the HiC data of the delta parB strain, indicates that the second diagonal present across the entire genome length. This is somewhat contradictory to the statements (line 335) that the arm cohesion is limited to approx. 2.0-2.2 Mbp. I agree that the arm cohesion in the core regions of the chromosome is more prominent, but it may not be absent towards the terminus. The time resolved data in Figure 1C show nicely that the arm cohesion is almost absent during hyphal growth, while it increases during sporulation. Might we face a mixture of chromosomal arrangements at time point 22h? My question goes into the direction of how fast the chromosomal rearrangements are. At the moment the text reads as if we look at a progressive dynamics of a chromosome during entrance into sporulation. I would rather see the data as a mix of differently organized chromosomes due to the different states there are in.

The authors write (lines 351-360) that at time point 13h the cells would be in a vegetative growth phase. Looking at the growth curves (Fig.1A), I do not agree that the cells at this time point are still in a log. growth phase.

Figure S3A (line 368): I would not say that there is no effect on growth rate when Par proteins are deleted. Deletion of Par Proteins influences later stages of the growth (e.g. sporulation).

Figure S3B: It seems that there is a mistake in the labels on the ParA data – currently it says 22h for all of them.

Line 395: What is the statistical basis for the notion that smc or hupS do “not significantly affect growth rate.”?

Fig. S5 Line 428/429. It is difficult to say that in the parB mutant the short range contacts are not affected. I do see a difference between wt and parB. This difference gets more pronounced in a smc mutant, though.

Fig. S9 is not referenced in order (referenced before S8).

Fig. S9B (lines 476-480): It is difficult to judge about HubS levels by one Western blot without loading control.

Line 853: Can the precise number of analyzed spores be indicated (approx. 300 sounds very vague).

Fig. 4 is not referenced in the main text. This should appear in the discussion I assume.

Reviewer #3 (Remarks to the Author):

The manuscript by Szafran et al. describes the global organization of the linear chromosome of the bacteria *Streptomyces venezuelae* during its entry to sporulation and the contributions of several nucleoid associated proteins to this organization (ParA, ParB, SMC and HupS). In this manuscript, the authors use various techniques including fluorescence microscopy, HiC and Chip-seq to better understand the remodeling of the chromosomes during this process and to assess the contributions of the previously mentioned proteins to these dynamic rearrangements.

Their work has multiple novel observations and would be of broad interest to readers studying chromosome dynamics in bacteria, chromosome organization in *Streptomyces* genera as well as SMC complex function across domains of life: i) they described for the first time the global organization of a linear bacterial chromosome, ii) they also described the global rearrangement during a differentiation of bacterial cells, iii) finally they described the implications of several structural proteins in these organization.

Overall, the manuscript is well-organized, substantially contributes to a better knowledge of bacterial chromosomes organization and merits publication.

However, specific queries with regards to biological replicates as well as some sections of the manuscript will need to be addressed before the manuscript is suitable for publication.

Major comments

1- First of all, it appears that, with the exception of the Δ parA mutant, there is no biological replicates shown for any of the HiC experiments. Given the resolution of the HiC matrixes and the variation observed for the Δ parA ones, it appears mandatory to show the biological replicates and

their reproducibility to support their claim especially concerning the implication of SMC and HupS on the local organization of the chromosomes. Several tools (like HiCrep) exist to compare contact matrix and to show correlation between them. Moreover, the Δ hupS matrix appears extremely noisy: is it an effect of the deletion or a problem of the library?

2- Concerning the treatment of HiC matrixes, I do not understand if bins are removed or set to 0 when HiC libraries are processed. Several areas in the matrix (like the one at 6.5 Mb) appear problematic and could disturb the analysis of the R/LTD domains. Maybe, the authors could annotate those weird bins and look for specific genetic features that could be responsible for this peculiar behavior.

3- As the authors take the whole mycelium to perform their HiC libraries, how they can be sure that the corresponding matrixes represent the organization of the differentiated cells? Are all the cells differentiating? Moreover, it appears intriguing to me that they were able to lyse spores using lysozyme. As far as I know spores are quite resistant and not easily breakable. A linked question: why using only Lysozyme for the HiC libraries and Lysozyme + Zircon beads for the Chip-seq experiments?

4- The scale bars of the HiC matrixes presented in all figures do not have quantitative value. It appears mandatory to provide those value and how the data are thresholded. Moreover, a table recapitulating the different libraries, the number of raw reads as well as the number of contacts kept in the different matrixes will be highly valuable.

5- Why there is no libraries performed between 17 and 22h of growth? It could have been very interesting to have the whole kinetics.

6- Line 316: Authors claim that chromosome compaction is initiated at 21h and increased till 25h. Is there a statistical test for this affirmation? it is hard to distinguish time point 21h,22h and 23h in the figure S1B.

7- Authors distinguished, based on PCA1/PCA2 analysis, three distinct chromosomes regions. Are those regions also detectable for each time points? Is it a conserved structure through time? are those regions dependent of the different tested structural proteins?

8- The authors say that the detected boundary of RTD and LTD correlates with lower transcriptional signal. However, if I am right, the RNA seq data used to detect this correlation was not done at the same development stage of *S. venezuelea*. Consequently, it appears quite problematic to claim this.

9- Is there a difference of nucleoid length between time points 13h and 17h? is it something measurable?

10- Is there an influence of ParB and ParA on the compaction of the nucleoid (i.e. nucleoid length) during the sporulation process?

11- I do not understand why the Chip-seq data from HupS-FLAG and SMC-FLAG were processed differently. In addition, why the Chip-seq for HupS was done in a Δ hupS strain while it was done in a WT strain for SMC?

12- What are the differences between the representation of the Chip-seq data for HupS in Fig. 3C and Fig. S8A? looking at the figure S8A it is difficult to imagine that HupS bound preferentially to RTD and LTD area.

Minor comments

1- Line 359: the reference to Fig. S2B do not seem to be the good one.

2- Fig. 1B the scale bar min/max is inverted.

3- The contact map for the Δ parB mutant at 17h show a clear barrier near the origin of replication. Can the author discuss this phenomenon? Is there a possibility that this barrier could be linked to a process linked to the differentiation or the compaction of the nucleoid?

4- It could be interesting if authors could indicate on Fig. 1B the different ParB binding sites. Are those sites all clustered around the origin of replication?

5- I do not know if citing a paper that is not already published is a good thing. Could the authors cite the BiorXiv paper instead (loy et al. Biorxiv - doi: <https://doi.org/10.1101/2020.12.09.415976>)

We are grateful to the reviewers for their thorough reviews of our manuscript and all valuable comments and corrections. We also appreciate them pointing out all the unclarities and inconsistencies. We have improved our manuscript and addressed all the comments below. We hope that we have clarified all the issues and that our manuscript is now acceptable for publication.

REVIEWER COMMENTS

Reviewer #1 (Remarks to the Author):

The manuscript by Szafran et al. is a very nice study on the structure of the linear Streptomyces venezuelae chromosome as the filamentous bacterium leaves vegetative growth, enters its developmental cycle and completes spore formation. The information is novel and the experiments have been carefully performed and well presented.

As a reader, the reviewer had difficulty really seeing the diffuse signal suggesting spatial proximity (lines 336-338) and contacts at the termini in Fig. 1B. The manuscript would be strengthened by spending a little more effort presenting this in the figure. Where is the diffuse signal? One needs to spend time looking at those types of plots and many people have not had much experience. What parts of the data in Figure 1 indicates that LTD and RTD are “organized”? This becomes one of the main conclusions of the study and would benefit from helping the reader early on in the manuscript.

To more clearly visualize the LTD and RTD in Fig. 1B, we marked their boundaries and added the following description to the figure legend:

“The boundaries of the LTD and RTD are marked directly on the Hi-C contact map with the orange dotted lines.”

To identify the existence of LTD/RTD, we used principal component analysis (PCA). In the revised version of the manuscript, we extended this analysis for different developmental stages of the wild-type strain as well as for other mutant strains described in the manuscript. Each analysis was performed with two experimental replicates (now Fig. S5). This analysis confirmed the HupS-dependent organization of the LTD and RTD.

We also added additional information to the Materials and Methods.

“To identify the LTD and RTD domain organization, principal component analysis (PCA) was performed using hicPCA (version 3.6+galaxy0). The generated bedgraph files were extracted and smoothed using the moving average, and the PCA1 scores for each 30 kb bin were plotted against the *S. venezuelae* chromosome position. The domain boundaries were identified based on the position of the X-axis intersection.”

To clearly indicate the range of the contacts along the secondary diagonal, we have now marked the contact range (Fig. 1B black dotted lines), indicating the signal dispersion close to the chromosomal termini, and we have expanded the interpretation of Fig. 1B as follows:

“Additionally, the Hi-C contact map showed some diffuse signals at the termini proximal fragments of the secondary diagonal, suggesting the existence of spatial proximity but also somewhat longer range contacts at the termini of the linear *S. venezuelae* chromosome (Fig. 1B and S3B).”

We also added this information to the figure legend:

“The contact range within the secondary diagonal axis is marked with black dots.”

Finally, we added the Hi-C matrix plotted using a heat-map colour pallet to the supplementary materials. Although this matrix is not appropriate to visualize contact frequencies, it is more suitable to show dispersed contacts at the chromosome termini (Fig. S3B). A gradient scale is broadly used to linearly correlate the numeric data with the colour intensity; thus, we decided to visualize normalized contact frequencies with a gradient scale. However, the disadvantage of the gradient scale is its discrimination of low-intensity reads, especially if directly confronted with strong signals (such as those in Fig. 1B, where very strong signals along the primary diagonal were shown).

2) *The structure of the Streptomyces chromosome is unknown to most other microbiologists that naturally think of a circular molecule with all unique coding sequences. Consider adding more of a description on the structure of linear chromosome in the introduction (either after line 72 or line 93). Information might be included is the fact that they have Terminal Inverted Repeats (TIR), and what the length of TIR is for S. venezuelae. That there are proteins covalently bound ends that are involved in replication of the ends of the lagging strand (“the end problem”) and are believed to interact for the ends of the linear chromosome to colocalize (line 94 and line 504). That there might be intermolecular end-to-end contacts before septation, in addition to intramolecular contacts as shown in the model of Figure 4.*

As suggested we added the following passage into the Introduction section:

“The termini of these linear chromosomes were shown to contain multiple short repeats forming regions called terminal inverted repeats (TIRs). TIRs are bound by TAPs (telomere-associated proteins) that are involved in the replication of the lagging strand at the end of chromosomal DNA and interact with other proteins named TPGs (terminal protein genes) to form a telomere-associated complex^{45,46}. The elimination of TAPs results in circularization of the chromosome^{47,48}. Interestingly, the length of TIRs varies for different Streptomyces species, ranging from less than a few hundred nucleotides (in *S. avermitilis*) to 1.4 Mb (in *S. ambofaciens*)⁴⁹”

3) *Continuing on the fact that the Streptomyces chromosome is linear and in reference to “Quantification of the oriC/ter ratio” (line 221), results line 304 and the legend to Figure 1. In E. coli, ter is a specific cis-acting DNA sequence. The ter sequences are specific locations opposite oriC to stall and dismantle replication forks. To the reviewer’s knowledge, the Streptomyces chromosome lacks ter sequences. The work in the manuscript uses the unique argG gene located more than 1 Mb from one end of the chromosome to determine when replication occurs. That should be more clearly and more correctly reflected in the manuscript. Similar issues are located in Figure S6A and legend.*

We corrected Fig. 1A according to the reviewer’s suggestion. The scheme of the chromosome was modified to clearly show the linearity of the Streptomyces replicating or nonreplicating chromosome. Moreover, to avoid misunderstandings, we have changed “the oriC/ter ratio” to “the oriC/arm ratio” throughout the manuscript. We also added, as supplementary information, the detailed localization of the oriC, gyrB and argG genomic positions (now Fig. S3A).

4) *Also continuing on the fact that the Streptomyces chromosome is linear, what is being shown with the cartoon of the chromosome in the inset of Figure 1A for the measurement of replication. It looks more like theta replication of a circular molecule and less like replication of a linear molecule. Most people would interpret the black dots on the circles as oriC (?) and not ends of chromosomes held together by terminal proteins (?).*

Please see the answer to the question #3

5) *The manuscript states that the SMC-FLAG protein is stable and the level is constant throughout the growth cycle (lines 438-439), but the stability of the FLAG-SMC seems to be affected in Figure S9C. The band at 12 h is larger than all of the other time points and there is a shadow of a band for full length in the 14-20 h time points. Should there be some concern about the tagged protein that actually accumulates in the cell and used for the ChIP-seq experiments?*

We have replaced the Western blot showing the FLAG-SMC level with a higher quality one (now also supplemented with an appropriate loading control), which clearly shows that the FLAG-SMC level is constant throughout development (now Fig. S10A).

6) *Line 132 – temperature of crosslinking?*

We added the following information: “[...]both steps performed at 30°C with shaking (180 rpm).”

7) *Line 140 – 10X or 1X NEB buffer?*

Added: “**10x NEB3**”

8) *Line 152 – Location of the sequencing facility?*

Added: “**(Plan-les-Ouates, Switzerland)**”

9) *Line 172 – Concentration and pH of the TES supplement?*

Corrected: [...] “**Trace Element Solution (TES, 0.2x culture volume)**” We also added a reference to Kieser et al. ‘Practical *Streptomyces* genetics’

10) *The Principle Component Analysis is not in the methods, but introduced in line 327 and Figure S2A. In the main text on line 327, only the abbreviation is used (PCA1/PCA2). The abbreviation should be spelled out at the first use.*

The information was added in the material and methods:

“To identify the LTD and RTD domain organization, principal component analysis (PCA) was performed using hicPCA (version 3.6+galaxy0). The generated bedgraph files were extracted and smoothed using the moving average, and the PCA1 scores for each 30 kb bin were plotted against the *S. venezuelae* chromosome position. The domain boundaries were identified based on the position of the X-axis intersection.”

Cosmetic comments:

11) *Line 21 – consider adding the word “central” to read “aligned within the central core region”.*

We prefer to use the phrase “core region” since the “central core region” may suggest that arm alignment is more limited.

12) *Line 24 – For the broad audience, add “HU-family” to read “Streptomyces-specific HU-family protein HupS”. The designation HU is more widely known than HupS. Consider a similar addition to line 389 as well.*

Done.

13) *Lines 35-36 contain some inadvertent duplicated text.*

Corrected.

14) Lines 63-64 refer to work in 3 organisms but all three references are for one organism (*B. subtilis*).

Corrected.

15) Line 84 references – The best reference for Z-ring spacing in sporogenic hyphae would be the original Schwedock et al. paper, or other *Streptomyces FtsZ* papers rather than the one on DynAB.

Corrected.

16) Line 309 – The extent of time of this pre-sporulation phase is probably 6-7 h instead of the listed 8 h? Using the manuscript time ranges listed in lines 305 and 309: 19-13=6 h and 21-14=7 h.

We have modified the text to make the length of the presporulation phase clearer and more consistent with what is shown in Fig. 1A. It now reads as follows:

“After 13-14 h of growth (under 5 ml culture conditions), we observed vegetative growth inhibition. Slowed hyphal growth correlated with a reduction in the oriC/arm ratio corresponding to decreased DNA replication (Fig. 1A and S3A), indicating entry into a developmental stage that we named the pre-sporulation phase, and which lasted until 22 h of growth (approximately 8 h). During this phase the DNA replication rate transiently increased (in 19-21 h), and this increased replication rate coincided with a slight increase in the optical density of the culture, which could be explained by the rapid growth of sporogenic hyphae (Fig. 1A).”

17) Line 324- Consider adding the growth time point (22h) in the sentence of the main text.

Added.

18) Line 349 – Consider adding the time 25h after “the final stage of spore maturation (25h), to be complete and consistent with the other times pointed out in the lines just preceding that.

Added.

19) Line 350 – Consider referring specifically to S1B rather than S1 to read “(Fig. 1A and S1B)”.

Corrected; now Fig. 1A and S4B.

20) Line 359 – The correct Figure to reference is Fig. S1B for nucleoid compaction.

Corrected. We modified slightly the Figure S1B (now Fig. S4B). See also the answer to the Reviewer #3.

21) Lines 376, 378, 381 and 383 – The reference should be to Fig 1D instead of Fig 1C.

Corrected.

22) Line 401 – Beginning here and at other places throughout, *hupS-FLAG* is correct, but the construct is *FLAG-smc*. Half of the times the manuscript used *FLAG-smc* and the other half *smc-FLAG*, including figures (e.g. Fig. 1B and S7B).

Corrected in several places.

23) Line 409 – The manuscript text should read “Z-ring spacing”. The cells would need to be stained for cell walls to be able to state “pre-spore compartments”.

Indeed, we have corrected this statement.

24) Line 429 – Should Fig. 1D be listed and not Fig. 1B?

Corrected.

25) Line 438 – Spelling of the word “remained”.

Corrected.

26) Line 475 – For clarity, consider changing “the low level” to either “a low level” or “an altered level” of HupS-FLAG.

Corrected.

27) Figure 1A – the scale bar in the microscopy image would be easier to see if it were wider and white on the black background. It is very difficult to see as the current thin red line. Are there white dashed lines on the Hi-C map in Figure 1B? It looks like there might be, but nothing is stated in the legend.

We corrected the scale bars.

The “white lines” in the Hi-C heat map are the result of data processing when some bins are omitted from the matrix as regions with very low or very high coverage (i.e., high number of BglII sites per 10000 bp, the presence of repeated sequences with a low number of unique sequences mapped to the genome). The filtering out of low/high coverage reads is a standard recommendation for Hi-C data correction. Thus, white line occurrence is common in Hi-C matrixes and is even more pronounced when the missing data are not compensated or corrected with statistical methods. In our processing pipeline, we did not use any bioinformatic tools for additional data smoothing.

28) Figure 1A legend – Line 829 stated that the *oriC/ter* ratio for 25 h was set as 1.0, but the plotted value is higher than 1.0.

We have changed the *oriC/ter* ratio to the *oriC/arm* ratio. The *oriC/arm* ratio after 26 h of growth was set to 1.0 in the experiment. We corrected this information.

29) Legend to Figure 2 – *hup-mCherry* and *mCherry-hup* are used back and forth in lines 859, 860, 864 and 865. The correct designation should be used consistently.

We used HupA-mCherry fusion. We have corrected this information in several places.

30) Legend to Figure S2 – Refer to Figure 1B in line 2 with the 22h data?

The reference to Fig. 1B has been added to the legend of the supplementary figure showing PCA1 analysis (current Fig. S5).

31) Legend to Figure S3 – Indicate growth was monitored at A600 for the Bioscreen C instrument? Are the 3 biological replicates all independent from the data in Figure 1D? or is one a repeat of the data in Figure 1D?

The legend of Fig. S2B (previously Fig. S3) has been amended. For the experimental replicates of the growth curve, the same strain was used as for the analysis presented in Fig. 1D.

32) *Figure S4D – The y axis label should be “Z-ring spacing” not “compartment length”.*

Corrected; now Fig. S7D.

33) *Figure S6 legend states that the ratio at 16h was set at 1.0, but the plot show that it is about 1.8.*

In this experiment, chromosomal DNA isolated from 26 h of culture under Hi-C experimental conditions (see Fig. 1A) served as our nonreplicating control and was set as 1.0. We clarified this information in the description in Fig. S6 (now Fig. S9A) as follows:

“The *oriC/arm* ratio was estimated as 1 in the culture (5 ml) growing for 26 h, corresponding to the appearance of spore chains “

34) *Figure S7C legend – Are the forward and reverse strands relative to the direction of replication?*

Based on Reviewer 3’s comment #8 (please see below), we have removed the discussion of transcriptional activity along the chromosome (previously Fig. S2), and we have decided to remove the previous Fig. S7C.

35) *Table S3 – Should “*oripUC*” be listed as a characteristic of pIJ773 and “*meijsca*” as a characteristic of pIJ10700?*

Corrected.

36) *Table S4 – There is a mistake and primers *Pgyr2_Rv* and *Pgarg3_Fd* have the same sequence listed. It also appears that primer *Psmc_promoter_Rv* is missing from the table.*

The primer sequences have been corrected.

37) *Supplementary information, line 1 under Construction of *smc* mutant strain – *apramycin* is misspelled.*

Corrected

Reviewer #2 (Remarks to the Author):

*Bacteria organize their chromosomes in a compacted structure, called nucleoid. This organization is essential to fit the large dimensions of the DNA strand into the tiny volume of the cell. Simultaneously, the compacted chromosome need to be replicated and accessible for transcription. Thus, the dynamic arrangement is organized in bacteria by a set of conserved proteins. Several nucleoid associated factors contribute to folding. In particular bacterial condensins (SMC) are required for loop formation and often for replichore cohesion. SMCs are loaded onto the chromosome by DNA-binding proteins, termed ParB. ParB binds to specific DNA sequences termed *parS* sites. These sites are usually scattered around the origin of replication. Streptomyces species display a fascinating life cycle. Vegetative hyphae contain multiple chromosomes and do not require regular cell division. Stationary cells develop into exospores. Spore formation requires coordinated DNA replication and cell division events, combined with precise chromosome segregation. Chromosome folding has been studied only in few model species.*

The authors of this manuscript analyzed the chromosomal organization of the Actinobacterium Streptomyces venezuelae. Like other Streptomyces this strain contains a linear chromosome and thus differs from most other bacteria. So far a detailed, HiC based analysis has not been performed in these bacteria. Therefore, this work is a valuable addition to the existing literature and adds to the completion of our understanding of bacterial chromosomal organization. The main findings are that the Streptomyces chromosome are organized by ParB-SMC action. In contrast to other bacteria, there is little arm cohesion during vegetative growth. However, during onset of sporulation the two chromosome arms get zipped up by SMC. SMC loading is dependent on ParB, similar to the situation of other bacteria. The terminal regions of the linear chromosome are folded into distinct domains, by action of the nucleoid associated protein HubS.

This paper contains an important set of data and is well written. The experiments are conducted with great care. Overall, this is a beautiful study. I have only a few questions that should be addressed in a revision.

1) *I interpret the strain list that all experiments were done with a strain carrying the ftsZ-ypet allele as sole copy of ftsZ. How does this strain grow in comparison to wild type? Earlier publications describe an effect on sporulation (delay). This might be discussed.*

In all studied strains, the *ftsZ-ypet* allele is an additional copy of the *ftsZ* gene. The same construct was used earlier as a marker of cell division in *S. venezuelae* (Donczew M et al., 2016, Schlimpert S. et al., 2017). We have confirmed that the additional copy of *ftsZ-ypet* does not disturb culture growth. To clarify this issue, we have added a comparison of the growth of the wild-type strain and the strain with an additional copy of *ftsZ* in Fig. S2A. We also added the following information to the Materials and Methods section:

“For the visualization of Z-rings, strains expressing *ftsZ-ypet* as the second copy of the *ftsZ* gene were used (the modification did not affect the growth rate, Fig. S2A).”

2) *It would be helpful if the authors can show where the parS sites are distributed in the S. ven. chromosome. In Fig. 1B the dotted line indicates the origin of replications. Does this mean the Streptomyces chromosome is organized from the oriC region? This would be an interesting difference, since other bacterial chromosomes are organized from their parS region. This aspect might be discussed.*

In *Streptomyces*, chromosome *parS* sites are clustered in the proximity of the *oriC* region. We have now added the scheme (Fig. S3B) showing their positioning and added the following information to the manuscript (Results):

“ParB has been shown to bind numerous *parS* scattered in the proximity of *oriC* (Fig. S3B), assembling into regularly spaced complexes that position *oriC* regions along the hyphae⁷⁵.”

3) *Figure 1: The marker frequency data are difficult so see in this arrangement. Consider restructuring.*

We have modified the scheme in Fig. 1A to make it more legible. Please also see the answer to question #3 of Reviewer #1.

4) *The differential maps based on the HiC data of the delta parB strain, indicates that the second diagonal present across the entire genome length. This is somewhat contradictory to the statements (line 335) that the arm cohesion is limited to approx.. 2.0-2.2 Mbp. I agree that the arm cohesion in*

the core regions of the chromosome is more prominent, but it may not be absent towards the terminus.

According to the suggestion, we have rephrased the text as follows:

“The secondary diagonal, which results from the interarm interactions, was prominent in the region of approximately 2.0-2.2 Mbp in each direction from the *oriC* region and severely diminished within terminal domains (Fig. 1B).”

5. The time resolved data in Figure 1C show nicely that the arm cohesion is almost absent during hyphal growth, while it increases during sporulation. Might we face a mixture of chromosomal arrangements at time point 22h? My question goes into the direction of how fast the chromosomal rearrangements are. At the moment the text reads as if we look at a progressive dynamics of a chromosome during entrance into sporulation. I would rather see the data as a mix of differently organized chromosomes due to the different states there are in.

We agree with the suggestion that the Hi-C maps show a mixture of chromosomal arrangements within the hyphae of developing strains and do not suggest the precise extent of cohesion. However, considering that the differentiation of *S. venezuelae* is fairly synchronized, e.g., cell division and chromosome compaction occur relatively synchronously within the culture, we believe that while we cannot estimate the timing of the process for a single chromosome, we can conclude that the average arm cohesion gradually increases. We added the following sentence to clarify this issue:

“Since the contact maps show various chromosomal arrangements within the hyphae of *S. venezuelae*, we cannot estimate the timing of the process for a single chromosome. However, taking into account synchronous *S. venezuelae* development, we can conclude that the average arm cohesion gradually increases during the presporulation phase.

6) The authors write (lines 351-360) that at time point 13h the cells would be in a vegetative growth phase. Looking at the growth curves (Fig.1A), I do not agree that the cells at this time point are still in a log. growth phase.

Streptomyces vegetative growth encompasses a log phase of growth and a stationary phase before the onset of sporulation. As we determined, none of the events related to sporulation (sporogenic hyphal growth, chromosome compaction, cell division) occurred before 17 hours of growth; therefore, we regard the 13th hour as the vegetative growth stage. To make it clearer we rephrased text as follows:

“To this end, we obtained normalized Hi-C contact maps for the wild-type strain during the vegetative stage at the entry into the presporulation phase when growth slows and DNA replication decreases (13 h) and during the presporulation phase when the chromosomes are still not compacted (15 and 17 h), as well as during spore maturation (25 h), which is when chromosome compaction reaches a maximum (Fig. 1A and S4B).”

7) Figure S3A (line 368): I would not say that there is no effect on growth rate when Par proteins are deleted. Deletion of Par Proteins influences later stages of the growth (e.g. sporulation).

We believe that the influence of ParA proteins on the overall rate of growth is negligible; however, it was shown that their elimination indeed influences the maturation of aerial hyphae (Donczew,

2016). Fig. S2B (earlier S3A), however, confirms that we are able to use the same time points for the analysis of chromosomes in different developmental stages.

We have clarified this as follows:

“Elimination of *S. venezuelae* segregation proteins disturbs aerial hyphal development and chromosome segregation³⁹ but does not affect the culture growth rate (Fig. S2B).”

8) *Figure S3B: It seems that there is a mistake in the labels on the ParA data – currently it says 22h for all of them.*

Figure S6B (earlier S3B) shows replicates of the analysis performed at the same time point (22h) for the *parA* deletion strain; it has also been clarified in the figure description. The replicates were shown to visualize slightly different Hi-C contact maps obtained for different experimental repeats.

9) *Line 395: What is the statistical basis for the notion that *smc* or *hubS* do “not significantly affect growth rate.”?*

To support this statement, we have now added the statistical analysis to the Fig. S7A (earlier Fig. S4A).

10) *Fig. S5 Line 428/429. It is difficult to say that in the *parB* mutant the short range contacts are not affected. I do see a difference between wt and *parB*. This difference gets more pronounced in a *smc* mutant, though.*

We corrected the statement as follows:

Previously: “By contrast, the elimination of *smc* resulted in a slight increase in short-range DNA contacts along the primary diagonal, whereas in the *parB* mutant, the short-range DNA contacts were not affected”

Now: **“In contrast, the elimination of *smc* resulted in an increase in short-range DNA contact frequency (<200 kb) along the primary diagonal, whereas in the *parB* mutant, the short-range DNA contacts were only slightly affected, suggesting an additional role of SMCs in chromosome organization (Fig. 1D, 3A and S8A). Indeed, while nucleoid compaction was previously noted to be affected by *parB* deletion³⁹, the measurements of the nucleoid area confirmed less chromosome decompaction in *parB* than in *smc* mutant strains (Fig. S8B).”**

11) *Fig. S9 is not referenced in order (referenced before S8).*

Corrected.

12. *Fig. S9B (lines 476-480): It is difficult to judge about *HubS* levels by one Western blot without loading control.*

We have now modified the supplementary figure (current Fig. S10B) to include the loading controls. While transcriptional analysis indicates an increase in *hupS* levels during the transition from vegetative growth to sporulation (Salerno et al., 2009, Bush et al., 2019 data, and our unpublished results), repeated Western blots indicate that the changes in *hupS* during the sporulation phase (12-16 h in ChIP-seq culture conditions) are negligible.

13) *Line 853: Can the precise number of analyzed spores be indicated (approx. 300 sounds very vague).*

We corrected the sentence indicating the exact number of analyzed spores (n = 300)

14) Fig. 4 is not referenced in the main text. This should appear in the discussion I assume.

The references to Fig. 4 were added in several places.

Reviewer #3 (Remarks to the Author):

The manuscript by Szafran et al. describes the global organization of the linear chromosome of the bacteria Streptomyces venezuelae during its entry to sporulation and the contributions of several nucleoid associated proteins to this organization (ParA, ParB, SMC and HupS). In this manuscript, the authors use various techniques including fluorescence microscopy, HiC and Chip-seq to better understand the remodeling of the chromosomes during this process and to assess the contributions of the previously mentioned proteins to these dynamic rearrangements.

Their work has multiple novel observations and would be of broad interest to readers studying chromosome dynamics in bacteria, chromosome organization in Streptomyces genera as well as SMC complex function across domains of life: i) they described for the first time the global organization of a linear bacterial chromosome, ii) they also described the global rearrangement during a differentiation of bacterial cells, iii) finally they described the implications of several structural proteins in these organization. Overall, the manuscript is well-organized, substantially contributes to a better knowledge of bacterial chromosomes organization and merits publication.

However, specific queries with regards to biological replicates as well as some sections of the manuscript will need to be addressed before the manuscript is suitable for publication.

Major comments

1) First of all, it appears that, with the exception of the $\Delta parA$ mutant, there is no biological replicates shown for any of the HiC experiments. Given the resolution of the HiC matrixes and the variation observed for the $\Delta parA$ ones, it appears mandatory to show the biological replicates and their reproducibility to support their claim especially concerning the implication of SMC and HupS on the local organization of the chromosomes. Several tools (like HiCrep) exist to compare contact matrix and to show correlation between them. Moreover, the $\Delta hupS$ matrix appears extremely noisy: is it an effect of the deletion or a problem of the library?

To compare the replicates of the Hi-C matrixes, as well as the particular time points or the analysed mutant strains, we used the *hicCorrelate* package. The results of pairwise, multiple comparisons are now shown as a heat diagram in a supplementary figure (Fig. S1A). Moreover, based on the Pearson correlation coefficient, the Hi-C matrixes were grouped due to their similarity, showing a correlation between particular Hi-C contact maps. Additional information was also added to the Materials and Methods section:

“The corrected matrixes served as the input files for the multiple pairwise matrix comparison using *hicCorrelate* (version 3.6+galaxy0). The Hi-C contact matrixes were compared and grouped based on the calculated Pearson correlation coefficients (Fig. S1B).”

We agree with the reviewer that *hupS* deletion matrixes show a high level of noise. Since these matrixes were independently repeated several times, always resulting in a similar overall noisy signal

and a decrease in the short-range contacts (Fig. S6C), we assume that the observed effect is the result of *hupS* deletion. Moreover, we also detected a noisy signal for the *hupS smc* double mutant but not for the *smc* single deletion (Fig. 3A), thus suggesting that the deletion of *hupS* is primarily responsible for the increase in random DNA contacts along the *Streptomyces* chromosome.

We have now added 3 independent replicates of the Hi-C contact maps (as well as their log2scale differential matrixes) constructed for the *hupS* deletion strain to the supplementary materials (Fig. S6C).

2) Concerning the treatment of HiC matrixes, I do not understand if bins are removed or set to 0 when HiC libraries are processed. Several areas in the matrix (like the one at 6.5 Mb) appear problematic and could disturb the analysis of the R/LTD domains. Maybe, the authors could annotate those weird bins and look for specific genetic features that could be responsible for this peculiar behavior.

To our knowledge, the Hi-C data correction algorithm (*hicCorrectMatrix* tool) removes inappropriate bins but does not set them to 0. Similarly, the bins with a coverage of 0 are also removed by the algorithm. A quote from the original tool manual is attached below:

“for the method to work correctly, bins with zero reads assigned to them should be removed as they can not be corrected. Also, bins with low number of reads should be removed, otherwise, during the correction step, the counts associated with those bins will be amplified (usually, zero and low coverage bins tend contain repetitive regions). Bins with extremely high number of reads can also be removed from the correction as they may represent copy number variations”

In our protocol, the threshold values to remove high and low data coverage (5.0 and -1.5, as stated in the manuscript) were estimated manually based on the close inspection of a generated diagnostic plot (we have added this information to the Materials and Methods section):

“The threshold values for the low and high data coverage were adjusted based on the diagnostic plot generated by the hicCorrectMatrix tool (version 3.3.1.0).”

Based on the quality check (*hicQuickQC* software), we determined that a high number of mappable pair reads were discarded due duplications (see Fig. S1B). This is a common cause of data loss during Hi-C data processing. However, we could not find any particular genetic feature that could explain why some regions of the *Streptomyces venezuelae* chromosome were overrepresented in the Hi-C library (and thus rejected at the stage of data correction).

Please also see the response to question #27 (Reviewer #1).

3) As the authors take the whole mycelium to perform their HiC libraries, how they can be sure that the corresponding matrixes represent the organization of the differentiated cells? Are all the cells differentiating? Moreover, it appears intriguing to me that they were able to lyse spores using lysozyme. As far as I know spores are quite resistant and not easily breakable. A linked question: why using only Lysozyme for the HiC libraries and Lysozyme + Zircon beads for the Chip-seq experiments?

Concerning the question of how synchronized the differentiation of the analysed culture is, please see the answer to Reviewer 2’s comment #5. Regarding the lysozyme used, we were not able to lyse the fully mature spores, and indeed, at the time point analysed (25 h), the spores were still immature in terms of cell envelope development. To avoid confusion, we have modified the text as follows:

“This stage ended when immature spores could be detected (25-26 h) (Fig. 1A and S4A).”

“To this end, we obtained normalized Hi-C contact maps for the wild-type strain during the vegetative stage at the entry into the presporulation phase when growth slows and DNA replication decreases (13 h) and during the presporulation phase when the chromosomes are still not compacted (15 and 17 h), as well as during spore maturation (25 h), which is when chromosome compaction reaches a maximum (Fig. 1A and S4B).”

“Finally, in the maturing spores (25 h), we observed almost complete arm alignment, similar to that detected at 22 h (Fig. 1C).”

Concerning the requirement to use lysozyme+zircon beads in the ChIP-seq experiments, this change was due to a large quantity of biomass obtained from 50 ml culture, in contrast to the 5 ml cultures typically used for Hi-C experiments. Moreover, in contrast to the ChIP-Seq experiment, in which a high amount of DNA was required for subsequent steps, for the Hi-C library, the required amount of DNA was only 1 µg; however, the DNA was expected to be of extremely good quality. The crucial step is to isolate intact chromosomal DNA, avoiding random DNA shearing, which could generate a high background signal. Thus, the use of zircon beads to disrupt *S. venezuelae* hyphae was omitted in the Hi-C protocol but not in the ChIP-Seq protocol.

4) The scale bars of the HiC matrixes presented in all figures do not have quantitative value. It appears mandatory to provide those value and how the data are thresholded. Moreover, a table recapitulating the different libraries, the number of raw reads as well as the number of contacts kept in the different matrixes will be highly valuable.

Since we used 0-1 normalization during data processing, we decided to manually adjust most of the Hi-C matrixes. The minimum value oscillated between 0.05 and 0.1, whereas the maximum value oscillated between 0.3 and 0.4. We added this information to the Materials and Methods.

“The threshold values (min/max) were manually adjusted for each Hi-C contact map independently, and the min and max values oscillated in the ranges of 0.05-0.1 and 0.3-0.4, respectively.”

To eliminate the subjective Hi-C data adjustment, we directly compared matrixes in the log2scale and showed them as differential Hi-C maps.

The only exception in the Hi-C analysis procedure described above is the contact map presented in Fig. 1B (wild-type 22 h). This map includes the high-frequency contacts between the neighbouring chromosomal loci along the primary diagonal. To highlight the long-range chromosomal contacts (standard protocol) in subsequent maps, we decided to skip the diagonal reads (the wild-type 22 h Hi-C matrix processed according to the standard protocol is shown in Fig. 3A and is based on the same input raw data as the Hi-C map in Fig. 1B).

We added additional information to the Materials and Methods section.

“skip diagonal counts: true (a standard pipeline) to highlight secondary diagonal contacts, or false to highlight primary diagonal contacts (for wild type 22 h; presented in Fig. 1B);”

According to the Reviewer’s suggestion we also added the detailed statistics of RAW and mappable reads to the supplement (Fig. S1B).

5) Why there is no libraries performed between 17 and 22h of growth? It could have been very interesting to have the whole kinetics.

While optimizing the Hi-C experiment (the preliminary experiment had a low sequencing depth and low final matrix quality), we collected data for the whole set of time points, including 20 h. Based on this preliminary data analysis, which showed little difference in the length of the secondary diagonals at 20 h of growth and at 22 h (see the figure below), we decided to omit this nonsignificant time point in subsequent high-depth library resequencing, and we focused particularly on 22 h of growth as the crucial time point.

6) Line 316: Authors claim that chromosome compaction is initiated at 21h and increased till 25h. Is there a statistical test for this affirmation? it is hard to distinguish time point 21h, 22h and 23h in the figure S1B.

We have modified Fig. S1B (now Fig. S4B) to include a statistical analysis of the chromosome compaction data using the Wilcoxon test, and accordingly, we have modified the statement commenting on this result as follows:

“At this time point, the spore chains entered the maturation phase; during this phase, chromosome compaction was shown to gradually increase, achieving significant condensation at 25 h (Fig. S4B),”

7) Authors distinguished, based on PCA1/PCA2 analysis, three distinct chromosomes regions. Are those regions also detectable for each time points? Is it a conserved structure through time? are those regions dependent of the different tested structural proteins?

To address this question, we have now added a detailed analysis of the PCA1 component, and each analysis included two experimental replicates (shown in Fig. S5). Based on the PCA1 analysis, we could clearly detect the LTD/RTD domains throughout *S. venezuelae* development as well as in the *smc*, *parA* and *parB* mutants but not in the *hupS* and *hupS/smc* mutants.

We added additional information in the material and methods section:

“To identify the LTD and RTD domain organization, principal component analysis (PCA) was performed using hicPCA (version 3.6+galaxy0). The generated bedgraph files were extracted and smoothed using the moving average, and the PCA1 scores for each 30 kb bin were plotted against the *S. venezuelae* chromosome position. The domain boundaries were identified based on the position of the X-axis intersection.”

8) The authors say that the detected boundary of RTD and LTD correlates with lower transcriptional signal. However, if I am right, the RNA seq data used to detect this correlation was not done at the same development stage of *S. venezuelae*. Consequently, it appears quite problematic to claim this.

Indeed, to avoid confusion since the co-submitted paper by Liroy et al. comprehensively analyses the relation between chromosomal domains and transcription, we have decided to remove this set of data.

9) *Is there a difference of nucleoid length between time points 13h and 17h? is it something measurable?*

In earlier stages of growth and differentiation than the 19th hour (now Fig. S4A and B), the chromosomes were not separated, and the nucleoid length was difficult to determine. As we now show in Fig. S4B, the measurements indicate that the difference in nucleoid compaction becomes significant only during spore maturation when the chromosomes are well separated.

10) *Is there an influence of ParB and ParA on the compaction of the nucleoid (i.e. nucleoid length) during the sporulation process?*

The influence of *parB* deletion on the compaction of chromosomes was noted earlier (Donczew 2016); here, we have compared the nucleoid area at the time of cell division in the wild type and *parB* and *smc* mutants. We noted chromosome decompaction in the *parB* mutant that was much less significant than that in the *smc* mutant. We described this observation as follows:

„Indeed, while nucleoid compaction was previously noted to be affected by *parB* deletion³⁹, the measurements of the nucleoid area confirmed less chromosome decompaction in *parB* than in *smc* mutant strains (Fig. S8B).”

11) *I do not understand why the Chip-seq data from HupS-FLAG and SMC-FLAG were processed differently. In addition, why the Chip-seq for HupS was done in a Δ hupS strain while it was done in a WT strain for SMC?*

The approach that we used in the analysis of HupS-FLAG binding was designed for proteins binding to specific regions on the chromosome. However, SMC binding is not specific, and this protein does not remain bound to a particular region of the chromosome. Therefore, we used a method that analyses the entire genome in bins of fixed size and looked for the enrichment of reads caused by SMC binding.

We have added the statement which clarify this as follows:

“However, quantification of the detected SMC binding sites along the chromosome (using an algorithms dedicated to non-specific DNA binders) showed that their frequency increased up to twofold in the chromosomal core region (Fig. 3B and S11A).”

12) *What are the differences between the representation of the Chip-seq data for HupS in Fig. 3C and Fig. S8A? looking at the figure S8A it is difficult to imagine that HupS bound preferentially to RTD and LTD area.*

Figure S8A (now Fig. S12A) shows only the normalized number of reads for HupS-FLAG and control strains and is meant to show the difference in read distribution between control strains and HupS-FLAG strains. Not all binding sites may be visible in this figure because of its resolution, especially since some HupS sites were located very close to each other. On the other hand, Fig. 3C shows the number of identified edgeR HupS binding sites along the chromosome (averaged over a 0.5 Mbp sliding window every 1000 bp).

Minor comments

1) *Line 359: the reference to Fig. S2B do not seem to be the good one.*

Corrected.

2) Fig. 1B the scale bar min/max is inverted.

Corrected.

3) The contact map for the $\Delta parB$ mutant at 17h show a clear barrier near the origin of replication. Can the author discuss this phenomenon? Is there a possibility that this barrier could be linked to a process linked to the differentiation or the compaction of the nucleoid?

We did not observe the chromosomal barrier for the additional replicate at the same time point or any other analysed time points. Here, we show all the replicates obtained for the *parB* deletion strain (Fig. S6A).

4) It could be interesting if authors could indicate on Fig. 1B the different ParB binding sites. Are those sites all clustered around the origin of replication?

Since all ParB-binding sites (*parS*) are clustered in the very close vicinity of the *oriC* region, showing them in Fig. 1B would decrease the clarity of the Hi-C maps. Thus, we have shown the *parS* site distribution in Fig. S3B, and we also comment on the position of *parS* sites in the text as follows:

“During *Streptomyces* sporogenic cell division, coincident with the time of observed for the folding of chromosomes with proximal arms, ParB has been shown to bind numerous *parS* scattered in the proximity of *oriC* (Fig. S3B), assembling into regularly spaced complexes that position *oriC* regions along the hyphae⁷⁵”

5) I do not know if citing a paper that is not already published is a good thing. Could the authors cite the BiorXiv paper instead (loy et al. Biorxiv - doi: <https://doi.org/10.1101/2020.12.09.415976>.)

The reference to Liroy et al. was added in several places.

REVIEWERS' COMMENTS

Reviewer #1 (Remarks to the Author):

The manuscript by Szafran et al. is a very nice study on the structure of the linear *Streptomyces venezuelae* chromosome as the filamentous bacterium leaves vegetative growth, enters its developmental cycle and completes spore formation. The information is novel and the experiments have been carefully performed and well presented.

I thank the authors for their thoughtful responses to the previous constructive criticism from the reviewers. In particular I appreciate the improvements in the figures for readers to "see" the data in the chromosome contact maps.

I have one major suggestion for strengthening the manuscript and several cosmetic comments.

1) One of the criticisms of the previous manuscript version was not giving the reader, who normally thinks about more common circular bacterial chromosomes, enough of an appreciate of the nature of the *Streptomyces* linear chromosome. The authors did respond and add more information in lines 91-99.

To enhance their improvements, consider making the information about the linear chromosome more accurate. Multiple short repeats are really "within what are called" terminal inverted repeats (lines 93-94). TIR is not just a series of short repeats as indicated by referencing the 1.4 Mb TIR of *S. ambofaciens* in line 99. Using the word "bound" in line 94 suggests to a reader that TAP is a standard DNA-binding protein when it is actually covalently linked to the 5' end of the DNA at each end of the chromosome. I had also expected the authors to explicitly state the length of TIR in *S. venezuelae* (line 99), the subject of this manuscript.

Pertaining to the issue of TIR, I had expected that the regions the authors defined here, as left and right terminal domains - LTD and RTD, were related to the length of the left and right TIR of the chromosome. This point could and should be made in the Discussion. This manuscript and the accompanying paper observed LTD and RTD in two species with very different lengths of chromosomal TIRs (lines 99 and 337). They are not one and the same. In line 333, the folded 1.5 and 2 Mb LTD and RTD are mainly unique sequences and very little of each is TIR. This issue may be helpful when discussing the pattern and distribution of HupS-binding sites?

Cosmetic Comments

2) If/when the manuscript is accepted, in the final uploaded version, the authors will need to scrutinize every reference. Something happened during the preparation of this revised version and many of the reference numbers point to incorrect references. This came to my attention when I noticed inadvertent duplicated text (lines 595-597 and 599-601). The duplicated text cites two different references (92 and 52) and neither are correct. I think the correct reference is 69. Then spot checking throughout the manuscript, there are many references that do not match. As an example, in line 193, the text is about ParB foci assembling regularly-spaced complexes in hyphae and reference 58 is about "Scaling read aligners to hundreds of threads on general-purpose processors".

3) Related to the duplicated text in above item 2, what is "application of repeated spore loading"? Is this clear from the text? How often? At what intervals?

4) Consider using "spatial organization" in line 26 of the abstract as organization is also in used in line 28 of the introduction. The word "rearrangement" draws to mind a physical recombination of DNA (a genetic rearrangement) and not changes to 3D spatial organization.

5) Line 81: This line should probably be reworded. "Followed by" should be "and", as these are simultaneous overlapping events (as stated in lines 121-122). Evenly spaced ParB foci begin segregation in syncytial sporogenic hyphae before synchronous cell division and chromosome compaction happens?

- 6) Line 125: The reader is incorrectly referred to Fig S2A, a growth curve, when the data being discussed is the DNA contact frequency.
- 7) Line 150: PCA1 should be spelled out as this is the first use of the term in the manuscript.
- 8) Lines 211 and 212: is "inset" the correct word. Are the "interpretation" panels actual insets in this figure?
- 9) Line 231: "Immunofluorescence" microscopy was not done. No antibodies were involved to detect the localization of fluorescent protein fusions.
- 10) Line 235: Consider removing "yellow arrowhead" here and only use it in the figure legend. The text cites three panels and only one has the yellow arrowhead.
- 11) Line 254: Refer to the parB data in panel 1D instead of wildtype 1B. The wildtype panel in 1B is already duplicated and shown in 3A.
- 12) Line 260: Consider rewording what comes across as an awkward double negative ("less chromosome decompaction").
- 13) Line 333: Inadvertent reference to protein "C-terminal domains" when referring to independently folded chromosomal regions.
- 14) Line 448: Measured at 1-h "increments" or "timepoints"?
- 15) Line 566: should 1 ng/ml read 1 ng/ul?
- 16) Line 584: What was the carbon source for minimal medium MM?
- 17) Line 774: "Practical" is misspelled in this reference.
- 18) Figure 2B: The legend box should have the blue data listed as "smc/ FLAG-smc".
- 19) Figure S1B: One of the data sets is listed as "par22_1" and probably should be listed as "parA22_1"? Control_22 is also listed as "no PFA", which is not defined in the legend.
- 20) Figure S3A: One yellow circle for oriC had a black outline while the other 2 do not. Why is the approximate symbol (~) used for each gene/locus location, which is listed to 3rd decimal point? Legend for S3B: list abbreviations LTD and RTD after the full words because the abbreviations are indicated by black dashed lines in the figure. Note that the location of gyrB gene (proxy for oriC) could be included in the expansion below figure S3B.
- 21) Figure S4B: The double-headed arrow is small, blurry and difficult to see in the cartoon on the right.
- 22) Figure S5: Consider changing the data order. The panel for smc could be put last so that the data for hupS is just above and smc just below the data for the hupS smc double mutant data. Consider revising the first line of the legend, which is not a sentence at present.
- 23) Figure S7A: the way that the growth rate for um/min was determined is not in the methods. Is the number in parentheses a standard deviation? The stated rates of ~5 um/min is 5 times the length of a typical rod shaped bacterial cell per minute?
- 24) Table S1: The genotypes of strains TMO18 and KP4F4 are incorrect (smc::FLAG-smc). The FLAG tagged version is at the phiBT1 att site as in TMO19. TMO19 is incorrect as having smc-FLAG instead of FLAG-smc.
- 25) Table S3: Plasmid pMS83 is not in the table.

26) Last section of supplementary information: In the section titled, "Construction of smc complemented with smc-FLAG" the FLAG tag is listed incorrectly in the title and 3 places in the text. It should read FLAG-smc.

Reviewer #2 (Remarks to the Author):

The authors of the revised manuscript "Spatial rearrangement of the *Streptomyces venezuelae* linear chromosome during sporogenic development" have now adequately addressed all my concerns. The paper is a wonderful study that will broaden our view on bacterial chromosome organization. Congratulations to all authors!

Reviewer #3 (Remarks to the Author):

Given the different remarks and comments, the revised version of the manuscript is satisfactory and suited for publication. The authors have well addressed the different comments and have performed the different analysis/experiments. This manuscript will be a nice publication allowing a better vision of chromosome organisation in bacteria with a linear chromosome.

We thank Reviewers for additional comments on our manuscript. Please see the detailed answers to comments below.

REVIEWERS' COMMENTS

Reviewer #1 (Remarks to the Author):

The manuscript by Szafran et al. is a very nice study on the structure of the linear *Streptomyces venezuelae* chromosome as the filamentous bacterium leaves vegetative growth, enters its developmental cycle and completes spore formation. The information is novel and the experiments have been carefully performed and well presented.

I thank the authors for their thoughtful responses to the previous constructive criticism from the reviewers. In particular I appreciate the improvements in the figures for readers to “see” the data in the chromosome contact maps.

I have one major suggestion for strengthening the manuscript and several cosmetic comments.

1) One of the criticisms of the previous manuscript version was not giving the reader, who normally thinks about more common circular bacterial chromosomes, enough of an appreciate of the nature of the *Streptomyces* linear chromosome. The authors did respond and add more information in lines 91-99.

To enhance their improvements, consider making the information about the linear chromosome more accurate. Multiple short repeats are really “within what are called” terminal inverted repeats (lines 93-94). TIR is not just a series of short repeats as indicated by referencing the 1.4 Mb TIR of *S. ambofaciens* in line 99. Using the word “bound” in line 94 suggests to a reader that TAP is a standard DNA-binding protein when it is actually covalently linked to the 5’ end of the DNA at each end of the chromosome. I had also expected the authors to explicitly state the length of TIR in *S. venezuelae* (line 99), the subject of this manuscript.

We thank Reviewer for this suggestion, now we have clarified the description of TIRs as follows:

“These linear chromosomes are flanked with terminal inverted repeats (TIRs) which encompass palindromes that form telomeric secondary structures. Replication of telomers is mediated by covalently bound telomere-associated complex of terminal proteins (TPs)^{45,46}. The elimination of TPs results in circularization of the chromosome^{47,48}. Interestingly, the length of TIRs varies for different *Streptomyces* species, ranging from less than a few hundred nucleotides (in *S. avermitilis*⁴⁹), to 200 kb (in *S. ambofaciens*⁵⁰) or even over 600 kb (in *S. collinus*⁵¹), while in *S. venezuelae* it was determined to be about 1.5 kb⁵².”

Pertaining to the issue of TIR, I had expected that the regions the authors defined here, as left and right terminal domains - LTD and RTD, were related to the length of the left and right

TIR of the chromosome. This point could and should be made in the Discussion. This manuscript and the accompanying paper observed LTD and RTD in two species with very different lengths of chromosomal TIRs (lines 99 and 337). They are not one and the same. In line 333, the folded 1.5 and 2 Mb LTD and RTD are mainly unique sequences and very little of each is TIR. This issue may be helpful when discussing the pattern and distribution of HupS-binding sites?

The length of LTD and RTD seems be not related to TIR length. Rather LTD and RTD encompass the regions defined earlier as chromosomal arms. We have added this explanation to Discussion as follows:

*“The independent folding of core and terminal domain corroborates with distinguished by genomic analyses regions of *Streptomyces* chromosome: core that contains housekeeping genes and exhibits high synteny and 1.5-2 Mb long arms with low evolutionary conservation.”*

Cosmetic Comments

2) If/when the manuscript is accepted, in the final uploaded version, the authors will need to scrutinize every reference. Something happened during the preparation of this revised version and many of the reference numbers point to incorrect references. This came to my attention when I noticed inadvertent duplicated text (lines 595-597 and 599-601). The duplicated text cites two different references (92 and 52) and neither are correct. I think the correct reference is 69. Then spot checking throughout the manuscript, there are many references that do not match. As an example, in line 193, the text is about ParB foci assembling regularly-spaced complexes in hyphae and reference 58 is about “Scaling read aligners to hundreds of threads on general-purpose processors”.

The references have been corrected. We have removed also the duplicated part of the Methods.

3) Related to the duplicated text in above item 2, what is “application of repeated spore loading”? Is this clear from the text? How often? At what intervals?

The description of the time lapse microscopy procedure has been expanded according to Reviewer’s and Editor’s suggestions.

4) Consider using “spatial organization” in line 26 of the abstract as organization is also in used in line 28 of the introduction. The word “rearrangement” draws to mind a physical recombination of DNA (a genetic rearrangement) and not changes to 3D spatial organization.

The abstract has been modified by Editor, however we decided to leave the word “rearrangement” in the other sections of the manuscript.

5) Line 81: This line should probably be reworded. “Followed by” should be “and”, as these are simultaneous overlapping events (as stated in lines 121-122). Evenly spaced ParB foci begin segregation in syncytial sporogenic hyphae before synchronous cell division and chromosome compaction happens?

Indeed, this sentence has been corrected.

6) Line 125: The reader is incorrectly referred to Fig S2A, a growth curve, when the data being discussed is the DNA contact frequency.

*We corrected the reference also adding the sentence : “First, using *ftsZ-ypet* strain and having confirmed that production FtsZ-Ypet fusion protein does not affect *S. venezuelae* growth (Fig. S2A) we determined the time points at which critical sporulation events took place (...)”*

7) Line 150: PCA1 should be spelled out as this is the first use of the term in the manuscript.

Corrected.

8) Lines 211 and 212: is “inset” the correct word. Are the “interpretation” panels actual insets in this figure?

Corrected. The term “inset” has been replaced with “scheme”.

9) Line 231: “Immunofluorescence” microscopy was not done. No antibodies were involved to detect the localization of fluorescent protein fusions.

Corrected.

10) Line 235: Consider removing “yellow arrowhead” here and only use it in the figure legend. The text cites three panels and only one has the yellow arrowhead.

Corrected.

11) Line 254: Refer to the *parB* data in panel 1D instead of wildtype 1B. The wildtype panel in 1B is already duplicated and shown in 3A.

The reference to Fig. 1B has been replaced with 1D reference as suggested.

*Indeed, the Hi-C data for the wild type strain are shown in both Fig. 1B and Fig. 3A, however we believe the data duplication for the wild type strain in Fig. 3A will be helpful for clearer interpretation of phenotypes observed in *smc* and/or *hupS* mutants. Moreover, wild type Hi-C matrix in Fig. 1B was processed a slightly different than the matrix showed in 3A for better representation of short-range contacts along the primary diagonal. These reads were omitted in other Hi-C matrices including the wild type in Fig. 3A. Thus, we prefer to show both wild type Hi-C matrices in the manuscript.*

12) Line 260: Consider rewording what comes across as an awkward double negative (“less chromosome decompaction”).

Corrected

13) Line 333: Inadvertent reference to protein “C-terminal domains” when referring to independently folded chromosomal regions.

Corrected

14) Line 448: Measured at 1-h “increments” or “timepoints”?

Corrected as “1-h intervals”

15) Line 566: should 1 ng/ml read 1 ng/ul?

Corrected.

16) Line 584: What was the carbon source for minimal medium MM?

1% mannitol. We have added this information in the Methods.

17) Line 774: “Practical” is misspelled in this reference.

18) Figure 2B: The legend box should have the blue data listed as “smc/ FLAG-smc”.

The complementation of smc deletion was done using pMS82-smc-FLAG construct (integration vector allowing production of SMC with C-terminal FLAG fusion)

(please see the answer to comment #24)

19) Figure S1B: One of the data sets is listed as “par22_1” and probably should be listed as “parA22_1”? Control_22 is also listed as “no PFA”, which is not defined in the legend.

Corrected. We have also added to the figure legend: “The wild type strain growing for 22 h and not treated with paraformaldehyde (no PFA, see Methods) served as a control.”

20) Figure S3A: One yellow circle for oriC had a black outline while the other 2 do not. Why is the approximate symbol (~) used for each gene/locus location, which is listed to 3rd decimal point? Legend for S3B: list abbreviations LTD and RTD after the full words because the abbreviations are indicated by black dashed lines in the figure. Note that the location of gyrB gene (proxy for oriC) could be included in the expansion below figure S3B.

Corrected according to the Reviewer’s suggestions.

21) Figure S4B: The double-headed arrow is small, blurry and difficult to see in the cartoon on the right.

Corrected.

22) Figure S5: Consider changing the data order. The panel for smc could be put last so that the data for hupS is just above and smc just below the data for the hupS smc double mutant data. Consider revising the first line of the legend, which is not a sentence at present.

We have changed the data order as suggested. Now the data order corresponds better with figures 1 and 3.

23) Figure S7A: the way that the growth rate for $\mu\text{m}/\text{min}$ was determined is not in the methods. Is the number in parentheses a standard deviation? The stated rates of $\sim 5 \mu\text{m}/\text{min}$ is 5 times the length of a typical rod shaped bacterial cell per minute?

We have added the information on the growth rate measurement and corrected the growth rate from “ $\mu\text{m}/\text{min}$ ” to “ $\mu\text{m}/\text{h}$ ”.

24) Table S1: The genotypes of strains TMO18 and KP4F4 are incorrect ($\text{smc}::\text{FLAG}-\text{smc}$). The FLAG tagged version is at the phiBT1 att site as in TMO19. TMO19 is incorrect as having $\text{smc}-\text{FLAG}$ instead of $\text{FLAG}-\text{smc}$.

The TM019 strain is the strain in which smc deletion is complemented with $p\text{MS83}-\text{smc}-\text{FLAG}$ construct, while TM017, TM018 and KP4F4 have insertion of FLAG sequence in the chromosomal locus of smc – delivering FLAG-SMC.

To confirm that insertion of FLAG in chromosomal locus of smc and production of FLAG-SMC by the strain used for ChIP-seq does not affect the spore size we included additional panel in supplementary Fig. S10B.

25) Table S3: Plasmid $p\text{MS83}$ is not in the table.

$p\text{MS83}$ plasmid has been added to the Table S3

26) Last section of supplementary information: In the section titled, “Construction of smc complemented with $\text{smc}-\text{FLAG}$ ” the FLAG tag is listed incorrectly in the title and 3 places in the text. It should read $\text{FLAG}-\text{smc}$.

(please see the answer to comment #24)

Reviewer #2 (Remarks to the Author):

The authors of the revised manuscript “Spatial rearrangement of the *Streptomyces venezuelae* linear chromosome during sporogenic development” have now adequately addressed all my concerns. The paper is a wonderful study that will broaden our view on bacterial chromosome organization. Congratulations to all authors!

Reviewer #3 (Remarks to the Author):

Given the different remarks and comments, the revised version of the manuscript is satisfactory and suited for publication. The authors have well addressed the different comments and have performed the different analysis/experiments. This manuscript will be a nice publication allowing a better vision of chromosome organisation in bacteria with a linear chromosome.